# Modelling the aerosol chemical composition of the tropopause over the Tibetan Plateau during the Asian summer monsoon

Jianzhong Ma[1], Christoph Brühl[2], Qianshan He[3], Benedikt Steil[2], Vlassis A. Karydis[4], Klaus Klingmüller[2], Holger Tost[5], Bin Chen[6], Yufang Jin[6], Ningwei Liu[1], Xiangde Xu[1], Peng Yan[7], Xiuji Zhou[1], Kamal Abdelrahman[8], Andrea Pozzer[2], Jos Lelieveld[2,9]

[1]State Key Laboratory of Severe Weather & CMA Key Laboratory of Atmospheric Chemistry, Chinese Academy of Meteorological Sciences, Beijing, 100081, China
[2]Atmospheric Chemistry Department, Max Planck Institute for Chemistry, P.O. Box 3060, Mainz, Germany
[3]Shanghai Meteorological Service, Shanghai, 201199, Germany
[4]Forschungszentrum Jülich, Institute for Energy and Climate Research, IEK-8, Jülich, Germany
[5]Institute for Atmospheric Physics, Johannes Gutenberg University Mainz, Mainz, Germany
[6]Department of Land, Air, and Water Resources, University of California, Davis, CA95616, USA
[7]CMA Meteorological Observation Centre, Beijing, 100081, China
[8]Geology and Geophysics Department, College of Science, King Saud University, Riyadh, Saudi Arabia
[9]Energy, Environment and Water Research Center, Cyprus Institute, 1645 Nicosia, Cyprus

*Correspondence to*: Jianzhong Ma (majz@cma.gov.cn)

**Abstract.** Enhanced aerosol abundance in the upper troposphere and lower stratosphere (UTLS) associated with the Asian summer monsoon (ASM), is referred to as the Asian Tropopause Aerosol Layer (ATAL). The chemical composition, microphysical properties and climate effects of aerosols in the ATAL have been the subject of discussion over the past decade. In this work, we use the ECHAM/MESSy Atmospheric Chemistry (EMAC) general circulation model at a relatively fine grid resolution (about 1.1×1.1 degrees) to numerically simulate the emissions, chemistry and transport of aerosols and their precursors in the UTLS within the ASM anticyclone during the years 2010–2012. We find a pronounced maximum of aerosol extinction in the UTLS over the Tibetan Plateau, which to a large extent is caused by mineral dust emitted from the northern Tibetan Plateau and slope areas, lofted to an altitude of at least 10 km, and accumulating within the anticyclonic circulation. We also find that the emissions and convection of ammonia in the central main body of the Tibetan Plateau make a great contribution to the enhancement of gasphase $NH_3$ in the UTLS over the Tibetan Plateau and ASM anticyclone region. Our simulations show that mineral dust, water soluble compounds, such as nitrate and sulfate, and associated liquid water dominate aerosol extinction in the UTLS within the ASM anticyclone. Due to shielding of high background sulfate concentrations outside the anticyclone from volcanoes, a relative minimum of aerosol extinction within the anticyclone in the lower stratosphere is simulated, being most pronounced in 2011 when the Nabro eruption occurred. In contrast to mineral dust and nitrate concentrations, sulfate increases with increasing altitude due to the larger volcano effects in the lower stratosphere compared to the upper troposphere. Our study indicates that the UTLS over the Tibetan Plateau can act as a well-defined conduit for natural and anthropogenic gases and aerosols into the stratosphere.

# 1 Introduction

The upper troposphere and lower stratosphere (UTLS) is a transition region between the troposphere and stratosphere, ranging about ±5 km around the tropopause (Gettelman et al., 2011), and the region plays an important role in the stratosphere-troposphere exchange (STE) and influences the chemistry of both the troposphere and stratosphere (Holton et al., 1995;Fueglistaler et al., 2009;Gettelman et al., 2011). Aerosols in the UTLS have large impacts on stratospheric ozone and global climate through heterogeneous chemistry and radiative forcing (Solomon, 1999;Solomon et al., 2007;Solomon et al., 2011). By analyzing the Cloud-Aerosol Lidar and Infrared Pathfinder Satellite Observation (CALIPSO) scattering ratio data, Vernier et al. (2011) found an Asian Tropopause Aerosol Layer (ATAL) associated with the Asian summer monsoon (ASM), i.e. an enhancement of background aerosol levels in the UTLS, between 13-18 km above sea level (a.s.l, hereafter all altitudes are referred to a.s.l. except when specified differently), over a widespread area of Asia (5-105 °E, 15-45 °N) including the Tibetan Plateau in July–August, for each year between 2006 and 2009. Thomason and Vernier (2013) confirmed the existence of the ATAL, beginning in 1999 with the Stratospheric Aerosol and Gas Experiment (SAGE II) satellite extinction coefficient data, while they stated that "there is no evidence of an ATAL in the SAGE II data prior to 1998".

The ASM anticyclone is a typical circulation pattern in the UTLS during the Northern Hemisphere summertime, and the deep convection that leads to the ASM anticyclone plays a key role in the transport of pollutants from the boundary layer to the UTLS and their trapping within the anticyclone (Gettelman et al., 2004;Randel and Park, 2006;Park et al., 2007;Park et al., 2008;Randel et al., 2010). Satellite observations have shown that the concentrations of tropospheric trace gases (including $H_2O$, CO, $CH_4$, NO, $NO_2$, $C_2H_6$, $C_2H_2$, $N_2O$, and HCN) are higher within than outside the anticyclone during the ASM period (Kar et al., 2004;Park et al., 2004;Li et al., 2005;Randel and Park, 2006;Park et al., 2007;Park et al., 2009;Xiong et al., 2009;Randel et al., 2010). Observations of trace gases like hydrogen cyanide (HCN), a tropospheric pollutant from biomass burning, reveals a combined influence of the ASM and human activities on chemical constituents in the UTLS region (Randel et al., 2010). Although the ATAL presence has been confirmed by measuring the aerosol optical properties, it remains a challenge to identify specific chemical components from satellites. General circulation models coupled with chemistry have been used to study the chemical composition and formation of the ATAL (Li et al., 2005;Fadnavis et al., 2013;Neely et al., 2014;Yu et al., 2015;Gu et al., 2016;Lelieveld et al., 2018).

Li et al. (2005) revealed the enhancement of anthropogenic aerosols within the ASM anticyclone by global chemical transport model (GEOS-Chem) simulation, and proposed that the ASM anticyclone could "trap" anthropogenic pollutants emitted and lofted from India and southwest China. Fadnavis et al. (2013)simulated persistent maxima in black carbon (BC), organic carbon (OC), sulfate and mineral dust aerosols in the UTLS within the ASM  anticyclone using an aerosol-chemistry-climate model (ECHAM5-HAMMOZ) and showed that this layer of UTLS aerosols  is originated from the pollution sources in the boundary layer. Neely et al. (2014) studied the ATAL and its possible origin using an aerosol microphys, chemistry and climate coupled model (WACCM3). They argued that "the ATAL is most likely due to

anthropogenic emissions, but its source cannot solely be attributed to emissions from Asia". According to Neely et al. (2014), $SO_2$ emissions from China and India make about a 30% contribution to the sulfate aerosol extinction in the ATAL during volcanically quiescent periods. Yu et al. (2015) explored the composition and optical properties of the ATAL with a climate model (CESM1) coupled by a sectional aerosol model and found that "the ATAL is mostly composed of mixed organics and

sulfates". Gu et al. (2016) performed model simulations with GEOS-Chem, and their simulations showed elevated nitrate, sulfate, ammonium, OC and BC in the UTLS over the Tibetan Plateau and ASM region as well. Gu et al. (2016) concluded that "nitrate aerosol is simulated to be of secondary importance near the surface but the most dominant aerosol species in the UTLS over the studied region". Above all, the models could consistently simulate an enhancement of aerosols in the UTLS within the ASM anticyclone, but there is disagreement among these model results with respect to which chemical

component(s) make a dominant contribution to and what are the most important source areas of the ATAL.

The Tibetan Plateau plays a critical role in the troposphere-to-stratosphere transport of water vapor and air pollutants throughout the ASM (Zhou et al., 1995;Fu et al., 2006;Lelieveld et al., 2007;Yang et al., 2014;Yu et al., 2017). Zhou et al. (1995) proposed that the Tibetan Plateau might be an important pathway for the transport of pollutants in eastern Asia from the lower troposphere to the stratosphere during summertime while exploring the causes of ozone valley over the Tibetan

Plateau. Subsequent model simulations and satellite data analyses confirmed this assumption and indicated that deep convection over the Tibetan Plateau and its southern slope is "a short circuit" for the transport of water vapor and polluted air to the global stratosphere (Li et al., 2005;Fu et al., 2006). The middle troposphere centered over the southern Tibetan Plateau acts as "a well-defined conduit", where strong convection lofts air parcels in the boundary layer, mostly from the plateau and India/southeast Asia, into the ASM anticyclone (Bergman et al., 2013). A recent model study indicated that the

ASM anticyclone serves as "an efficient smokestack", venting aerosols with a substantial amount of organic and sulfates to the UTLS (Yu et al., 2017). While most studies have focused on the transport of pollutants originating from south and southeastern Asia to the UTLS over the Tibetan Plateau and within the ASM anticyclone, the influence of natural sources such as mineral dust and their interactions with pollution on the formation of ATAL has not been well addressed. The Cloud-Aerosol Lidar and Infrared Pathfinder Satellite Observations (CALIPSO) and Multi-angle Imaging SpectroRadiometer

(MISR) satellites detected the summertime dust aerosol plumes over the northwestern Tibet, most probably originating from the Taklamakan desert and lofted from the surface to an altitude of about 9 km around the source region (Huang et al., 2007;Huang et al., 2008;Xia et al., 2008). Further analysis of the CALIPSO and MISR satellite data indicated that "aerosols appear to be more easily transported to the main body of the Tibetan Plateau across the northern edge rather than the southern edge" and "dust is found to be the most prominent aerosol type on the Tibetan Plateau" (Xu et al., 2015). A 3-day

case study using the WRF mesoscale model coupled with a dust module showed that deep convection in early summer over the Tibetan Plateau can inject dust aerosols into the stratosphere (Yang et al., 2014). The contribution of these dust plumes to the total aerosols in the UTLS over the Tibetan Plateau and associated transport and transformation processes needs to be thoroughly investigated.

Volcanic eruptions are a large source of aerosols in the UTLS and can influence the global climate significantly (Robock, 2000;Vernier et al., 2009;Solomon et al., 2011;Bourassa et al., 2012;Zhuo et al., 2014). From the global-scale viewpoint, the effects of volcanic eruptions on stratospheric aerosols mainly arise not from the injected ash but from the secondarily formed aerosols in the plumes; the former is composed of large size particles that are rapidly removed by gravitational sedimentation,

while the latter are composed largely of dilute sulfuric acid droplets formed by the oxidation of injected $SO_2$ and have a long residence time in the atmosphere, typically one year or more in the stratosphere (Junge et al., 1961;Brock et al., 1995;Solomon et al., 2011). The Nabro volcano, located at the border between Eritrea and Ethiopia, northeastern Africa (13.37° N, 41.70° E, 2218 m a.s.l.), erupted on 12-13 June 2011, injecting approximately 1.3-2.0 Tg of $SO_2$ into the UTLS; the $SO_2$ and aerosol plumes with little amounts of ash were detected by different instruments onboard several satellites in a

few weeks after the eruption (Bourassa et al., 2012;Clarisse et al., 2012;Bourassa et al., 2013;Fromm et al., 2013;Theys et al., 2013;Vernier et al., 2013;Clarisse et al., 2014;Fairlie et al., 2014). The eruption of Mt. Nabro in June 2011 has been the largest single injection of $SO_2$ to the UTLS after the eruption of Mt. Pinatubo in June 1991 unitl now, and the dispersion of its plume to East Asia provides an actual case for examining the combined effects of the volcanic eruption and ASM anticyclone on the UTLS aerosols over the Tibetan Plateau and the ATAL (Fairlie et al., 2014).

In the present study, we investigate the chemical composition and source regions of aerosols in the UTLS over the Tibetan Plateau during the ASM, using the ECHAM/MESSy Atmospheric Chemistry (EMAC) model (Jöckel et al., 2006;Jöckel et al., 2010). The adopted model resolution is T106L90, which is finer than that typically used in previous studies (e.g., T42L31, T42L90 or T63L90), allowing the model to reproduce the processes affected by the topography more realistically. Our model simulations cover the period from January 2010 to December 2012, for which $SO_2$ plumes from

twenty-seven volcanic eruptions, including the Nabro eruption, had been detected by satellite and were implemented into the model. In addition, all other natural and anthropogenic sources of gases and aerosols, such as on-line calculated dust emissions, are taken into account. Comprehensive aerosol microphysical and gas/aerosol partitioning processes as well as a full chemical mechanism for both the troposphere and stratosphere implemented in EMAC are included in our simulation. Our study focuses on the seasonal mean spatial distributions of aerosol chemical components and associated optical

properties in the UTLS during the ASM in the three simulation years. A detailed description of the model settings is given in Sect. 2, and the results and discussion are presented in Sect. 3. A summary of the main findings and conclusions is given in Sect. 4.

## 2 Model description and setup

The EMAC model is a chemistry-climate model that combines the 5th generation European Centre – Hamburg general

circulation model (ECHAM5) (Roeckner et al., 2006) with the Modular Earth Submodel System (MESSy) Atmospheric Chemistry system (Jöckel et al., 2006;Jöckel et al., 2010) to simulate and predict atmospheric processes from the troposphere to middle atmosphere and their interactions with land and oceans. For this study, we used version 2.52 of EMAC, which includes the MESSy submodels describing various chemical, physical and dynamical processes in detail (Jöckel et al.,

2005;Jöckel et al., 2010;Jöckel et al., 2016). EMAC has been extensively used at a range of spatial resolutions and evaluated against site, aircraft and satellite measurements of trace gases and aerosols in both the troposphere and stratosphere (Jöckel et al., 2006;Lelieveld et al., 2007;Pozzer et al., 2007;Pringle et al., 2010;Tost et al., 2010;Astitha et al., 2012;Brühl et al., 2012;Pozzer et al., 2012;Brühl et al., 2015;Eichinger et al., 2015;Pozzer et al., 2015;Jöckel et al., 2016;Tsimpidi et al.,

2016;Abdelkader et al., 2017;Gottschaldt et al., 2017;Karydis et al., 2017;Bacer et al., 2018;Brühl et al., 2018;Gottschaldt et al., 2018;Khosrawi et al., 2018;Klingmüller et al., 2018;Lelieveld et al., 2018). The model spectral resolution used in this study is T106L90, which corresponds to a horizontal grid resolution of approximately $1.125^{o} \times 1.125^{o}$ and 90 vertical layers extending from the surface to an altitude of 0.01 hPa (~80 km) with a vertical resolution of about 500m in the tropopause region. In this study, the model simulation was performed for the years 2010–2012. One year spin-up simulation was

performed followed by three years of normal simulation for the period 2010−2012, and results from the latter are presented and discussed below.

EMAC simulates chemical reactions in the gas-phase online through the Module Efficiently Calculating the Chemistry of the Atmosphere (MECCA) submodel (Sander et al., 2011). MECCA calculates the concentration of a large number of gaseous species and radicals, including various gaseous precursors of aerosols, such as $SO_2$, $NH_3$, nitrogen oxides ($NO_x \equiv$

$NO + NO_2$), volatile organic compounds (VOCs), and dimethyl sulfide (DMS), and major oxidant species like OH, $O_3$, $H_2O_2$, and $NO_3$. Photolysis rates for the troposphere up to the mesosphere are calculated by the JVAL submodel (Jöckel et al., 2006), which considers absorption and scattering by gases, aerosols and clouds in a delta-two-stream method. The chemical mechanism used in this study is primarily based on stratospheric chemistry used by Brühl et al. (2015) plus VOC chemistry reported by Taraborrelli et al. (2012).

Aerosol microphysics and conversion between gaseous and aerosol specieses conversion are treated by the Global Modal-aerosol eXtension (GMXe) module (Pringle et al., 2010), which uses 7 interacting lognormal modes (M7) to describe the size distributions of aerosols in the nucleation mode and hydrophilic and hydrophobic Aitken, accumulation and coarse modes. The aerosol composition is uniform in size within each mode (internally mixed), while it can vary from one mode to another (externally mixed). Each mode is represented by the number concentration, the number mean radius and its

geometric standard deviation (Pringle et al., 2010).

Within the GMXe module, the nucleation schemes of Vehkamaki et al. (2002) was used, which calculates the nucleation of new particles as a function of temperature (for a range of 190 K< T <305.15 K), relative humidity (for a range of 0.01%< RH <100%) and the concentration of sulfuric acid ($H_2SO_4$). The primary OC aerosol is assumed to be emitted by 65% as hydrophilic and 35% as hydrophobic (Pringle et al., 2010). Following Pringle et al. (2010) and Pozzer et al. (2012),

secondary OC particles are directly emitted as primary OC. The organic aerosol formation and chemical aging can be calculated by the ORACLE submodel in EMAC (Tsimpidi et al., 2014). However, in this work the partitioning of secondary organic compound between the gaseous and aerosol phases is not considered, and the ORACLE module was not used in the simulation. The aging that leads the conversion of hydrophobic BC and OC to hydrophilic ones is realized by the collisions

with hydrophilic particles and the condensation of inorganic gases such as sulfuric acid (Pringle et al., 2010 and references therein). Inorganic aerosol chemistry is simulated by the ISORROPIA-II thermodynamic equilibrium model (Fountoukis and Nenes, 2007), with latest version introduced by Capps et al. (2012). ISORROPIA-II calculates the gas/liquid/solid equilibrium partitioning of the $Na^+$, $K^+$, $Ca^{2+}$, $Mg^{2+}$, $NH_4^+$, $SO_4^{2-}$, $NO_3^-$, $Cl^-$, and $H_2O$ aerosol system. Calcium, potassium, sodium, and magnesium are treated as chemically active crustal species existing in the form of 14 mineral salts ($Ca(NO_3)_2$, $CaCl_2$, $CaSO_4$, $KHSO_4$, $K_2SO_4$, $KNO_3$, $KCl$, $MgSO_4$, $Mg(NO_3)_2$, $MgCl_2$, $NaHSO_4$, $Na_2SO_4$, $NaNO_3$, $NaCl$) and four ions in the aqueous phase ($Ca^{2+}$, $K^+$, $Mg^{2+}$, $Na^+$). To account for kinetic limitations, ISORROPIA-II calculates the gas/aerosol partitioning by two stages (Pringle et al., 2010). First, assuming that condasation is diffusion limited it calculates the amount of the gas-phase species that is able to kinetically condense onto the aerosol phase within the model time step,. Second, the derived mass is redistributed between the gas and the aerosol phase, with an assumption of instant equilibrium between the two phases. In ISORROPIA-II aerosol liquid water (ALW) of the simulated aerosol compounds is determined based on the Zdanovskii-Stokes-Robinson (ZSR) relationship (Stokes and Robinson, 1996), with the uptake limited by a grid box mean relative humidity of 95%.

Heterogeneous reactions of species on aerosol surfaces are simulated using the MECCA_KHET submodel (Jöckel et al., 2010). MECCA_KHET calculates the tropospheric heterogeneous reaction rates as mass transfer rates based on the aerosol surface density from GMXe (Jöckel et al., 2010). For the stratosphere the Multiphase Stratospheric Box Model (MSBM), which is coupled to MECCA_KHET, takes the input from GMXe to calculate the heterogeneous reaction rates on polar stratospheric cloud (PSC) particles and on stratospheric aerosols (Jöckel et al., 2010;Kirner et al., 2011).

Aerosol extinction is calculated online by the AEROPT submodel based on Mie theory using pre-calculated look-up tables for the six aerosol components, including black carbon (BC), organic compounds (OC), mineral dust (DU), sea salt (sea spray), water-soluble species (WASO), i.e., all other water soluble inorganic species (e.g., $NH_4^+$, $HSO_4^-$, $SO_4^{2-}$, $NO_3^-$), and ALW in the Aitken, accumulation, and coarse modes.

The uptake of $SO_2$, $HNO_3$ and $NH_3$ and the aqueous-phase oxidation of $SO_2$ by $H_2O_2$ and $O_3$ in clouds are calculated by the SCAV submodel (Tost et al., 2006a;Tost et al., 2007a). The scavenging of gases and aerosols by wet deposition is calculated by the SCAV submodel (Tost et al., 2006a), and their removal through dry deposition is calculated by the DRYDEP submodel (Kerkweg et al., 2006). The SEDI submodel with a first-order trapezoid scheme is used to calculate the sedimentation of aerosols (Kerkweg et al., 2006).

We used an emission inventory of the Representative Concentration Pathways scenario 8.5 (RCP8.5) , namely MACCity emission inventory, for fossil fuel combustion and biomass burning emissions in our simulation (see Jöckel et al. (2016) and references therein). The RCP8.5 global emission inventory has a horizontal grid resolution of 0.5 °× 0.5 °at monthly intervals and vertical distributions as described in Pozzer et al. (2009), and it is a reasonable choice for anthropogenic emissions over the period from 2000 to 2010 (Granier et al., 2011;Pozzer et al., 2015). The monthly cycle of the RCP8.5 emissions in 2010 are used in this study. The natural emissions of non-methane hydrocarbons (NMHCs) and $NH_3$ are based on the Global Emissions InitiAtive (GEIA) database (Bouwman et al., 1997). The NO emissions from soils are calculated online according

to the algorithm of Yienger and Levy II (1995). Carbonyl sulfide (OCS) (Montzka et al., 2007) and other long-lived source gases are constrained by observed surface volume mixing ratios (in monthly zonal average value), as done in Brühl et al. (2015). The oceanic dimethyl sulfide (DMS) emissions are calculated online by the AIRSEA submodel (Pozzer et al., 2006). Sea spray aerosol emissions are estimated offline using the monthly emission data set of AEROCOM (Dentener et al., 2006), with a composition fraction of 55% $Cl^-$, 30.6% $Na^+$, 7.7% $SO_4^{2-}$, 3.7% $Mg^{2+}$, 1.2% $Ca^{2+}$, and 1.1% $K^+$ (Seinfeld and Pandis, 2006).

Convection is parameterized by the submodel CONVECT, in which the Tiedtke (1989) scheme with modifications by Nordeng (1994) is set as default (Tost et al., 2006b) and used in this study, This scheme was developed for resolution T63. As discussed in Brühl et al. (2018), it overestimates vertical transport in the UTLS for T42 and underestimates it for T106. The $NO_x$ emissions from lightning activity are calculated online using the submodel LNOX (Tost et al., 2007b). In this study we apply the parameterization developed by Grewe et al. (2001), which links the flash frequency to the updraft velocity. Here flash frequency obtained by this parameterization is scaled by a factor of 0.0695 for T106L90 simulations. In this study global emissions of $NO_x$ from lightning are simulated to be 7.9, 6.7 and 6.3 Tg (N) $a^{-1}$ for the years 2020, 2011 and 2012, respectively, falling into the range of 2–8 Tg (N) $a^{-1}$ suggested by Schumann and Huntrieser (2007). The parameterisation schemes for convection (Tiedtke, 1989; Nordeng, 1994; Tost et al., 2006b) and lightning (Grewe et al., 2001) used in this study have been evaluated with simulations at coarser resolution (Tost et al., 2007b;Lopez, 2016;Gottschaldt et al., 2018) and the agreement is particularly noticeable for the ASM anticyclone region (Gottschaldt et al., 2018).

Dust emissions are calculated online with an advanced dust flux scheme, which was developed by Astitha et al. (2012) and updated recently by Klingmüller et al. (2018). The emission scheme combines meteorological parameters, which are derived online from the EMAC model, with soil properties including the soil clay fraction, the rooting depth and the vegetation area index, which are prescribed at monthly intervals based on the latest satellite observations (Astitha et al., 2012;Klingmüller et al., 2018). The dust particles turn to be mobilized in the air when the wind friction velocity exceeds a threshold value, which is determined by the soil texture classification and soil particle size distribution. Considering the regional differences in the soil particle size distribution, the emission scheme does provide an explicit variant (DU_Astitha2) based on soil characteristics of each model grid cell (Astitha et al., 2012). In the present study, we chose an alternative variant (DU_Astitha1), which assumes that the dust particles emitted from different soils have the same size originally. Compared to DU_Astitha2, the DU_Astitha1 scheme can achieve competitive results with reduced complexity and has been used in recent studies as well (Abdelkader et al., 2015;Abdelkader et al., 2017;Klingmüller et al., 2018). We used the DU_Astitha1 in combination with new land cover classification, vegetation, clay fraction and topography data set, which has been implemented by Klingmüller et al. (2018) based on up-to-date observations and can well represent the rapid changes of deserts and semi-arid regions in recent decades. Emissions of individual crustal species are calculated as a constant fraction of general mineral dust emissions, and this fraction can vary with the different dust source regions (Karydis et al., 2016). In this study, spatially uniform mineral dust composition is assumed, with a crustal species fraction of 1.2% $Na^+$, 1.5% $K^+$, 2.4% $Ca^{2+}$, and 0.9% $Mg^{2+}$ in the dust (Sposito, 1989).

As in the work of Brühl et al. (2018), the volcanic $SO_2$ plumes detected by the Michelson Interferometer for Passive Atmospheric Sounding (MIPAS) satellite were used to account for the volcanic $SO_2$ emissions in the simulation of this study. MIPAS (Fischer et al., 2008) is a limb sounder on board the Envisat sun-synchronous polar orbiting satellite, which was launched on 1 March 2002 and lost contact with the ground on 8 April 2012. Global distributions of $SO_2$ ranging from the upper troposphere to 20 km altitude have been retrieved from the thermal radiation (685–2410 $cm^{-1}$) emitted by the atmosphere as measured by MIPAS, and 3-D fields of volcanic $SO_2$ plumes in the UTLS with a temporal resolution of 5 days were derived (Höpfner et al., 2015). Since MIPAS cannot detect very fresh plumes as the field of view is obscured by ash, in most cases two to six subsequent 5-day periods are used for extrapolation of the initial spatial $SO_2$ distribution. As shown in Fig. 1a, there were about 27 explosive volcanic eruption events occurring over the period of January 2010 through March 2012. The highest $SO_2$ mixing ratio of >1 nmol $mol^{-1}$ (or ppbv) was detected in the Nabro eruption plume. As can be seen in Fig. 1b, peak ejections generally occurred at altitudes of about 16-17 km, i.e., around the thermal tropopause (defined by the World Meteorological Organization (WMO)). The volume mixing ratios of the volcanic $SO_2$ plumes from MIPAS were added to the simulated background $SO_2$ mixing ratios at the eruption time by using the restart technique implemented in EMAC. Due to limitations of MIPAS data, no volcano eruption is considered in our simulation for the period after March 2012. There were two large major volcanic eruptions occurring in 2012, namely Soputan (124.73 °E, 1.11 °N) on 18 September 2012 and Copahue (288.8 °E, 37.86 °S) on 22 December 2012, with the amounts of ejected $SO_2$ being 1-2 orders of magnitude smaller than that of Nabro, as reported by Mills et al. (2016). These two volcanic eruptions are not included for the simulations of this study. The biases of the MIPAS $SO_2$ data were estimated to be within 50 pmol $mol^{-1}$ (or pptv) in case of the volcanically enhanced concentrations (Höpfner et al., 2015). Detailed information on the name, time, location and the amount of injected $SO_2$ for individual eruption can be found in the work of Bingen et al. (2017) and (Brühl et al., 2018) and references therein.

In the simulation, the meteorology was nudged by Newtonian relaxation towards the European Centre for Medium-Range Weather Forecasts (ECMWF) operational re-analysis data for the years 2010–2012. Temperature, vorticity, divergence and surface pressure are nudged towards the realistic meteorological conditions , and the nudging weights are chosen such that the mesosphere and upper stratosphere and the boundary layer are not directly influenced (Lelieveld et al., 2007). The nudging was exerted with maximum weights at the model levels from 37 (~ 10 hPa) to 93 (~706 hPa), leaving the highest thirty (upper middle atmosphere) and the lowest three (boundary layer) model levels free (apart from surface pressure), as used by Jöckel et al. (2016). The chemical initial conditions for this model study were provided by the results from a previous simulation of EMAC T42L90 (Brühl et al., 2015) for all gases and aerosols, except for NMHCs which were from an EMAC T106L31 simulation (Pozzer et al., 2012). Pozzer et al. (2012) used the same modules (e.g., GMXe and ISORROPIA-II for aerosols) as in this study but different emission data (e.g., dust emissions from AEROCOM) in the T106L31 simulations of tropospheric aerosols for the years 2005−2008. By comparing the model output with observations from different measurement networks and satellite remote sensing instruments, they found that the main spatial and temporal atmospheric distribution of sulfate, ammonium and nitrate aerosols were well reproduced in general, but there was an

underestimation of aeolian dust emissions in the dust outflow regions (Pozzer et al., 2012). Klingmüller et al. (2018) develop an advanced dust emission scheme, with the updated land cover classification, the inclusion of the topography factor and the modification of the sandblasting efficiency function. They performed the T106L31 simulations for the year 2011; by comparing with the aerosol optical depth (AOD), dust concentrations and deposition results from various observational

platforms, they concluded that the update significantly improves agreement with the observations (Klingmüller et al., 2018). With respect to aerosol and stratospheric chemistry, the model setup in this study is similar to those used in Brühl et al. (2015, 2018), the latter of which (Brühl et al.,2018) used the T106L90 resolution for a 1-year sensitivity test.

## 3 Results and Discussion

### 3.1 General features

The ASM circulation is characterized by cyclonic flow and convergence in the lower troposphere, and a strong anticyclone and divergence in the UTLS (Krishnamurti and Bhalme, 1976); its structure is primarily a response to diabatic heating associated with deep convection (Hoskins and Rodwell, 1995). Known as the "sensible heat pump", the Tibetan Plateau modifies monsoon circulation (Wu and Zhang, 1998) and even modulates large-scale atmospheric circulations over the Northern Hemisphere (Zhao and Chen, 2001;Zhou et al., 2009). The strength and position of the ASM anticyclone can be

represented by different parameters, such as the circulation stream function (Randel and Park, 2006;Park et al., 2007), geopotential height (Bergman et al., 2013;Pan et al., 2016;Li et al., 2017) or potential vorticity (Garny and Randel, 2013;Ploeger et al., 2015). In this study we use the pressure deviation (d$P$) as a measure, defined as the difference of pressure, $P$, at each model grid relative to their regionally averaged value over 20–140 °E and 10–45 °N, $P_{avg}$, at the same altitude. We used the d$P$ value of 1 hPa (and 1.3 hPa) to plot the contour lines. This selection is arbitrary, but it appears to

work well as demonstrated below. Note that the ASM circulation is not steady but exhibits substantial intra-seasonal variability, with northward propagation during the onset phase of the monsoon and westward propagating features during the mature phase (Krishnamurti and Ardanuy, 1980;Yasunari, 1981). There is eddy shedding to the east and to the west of the ASM anticyclone. Here we focus on the seasonal averaged characteristics associated with the ASM.

Figure 2 shows EMAC simulated horizontal distributions of d$P$ overlaid by the horizontal wind field, O$_3$ and CO,

averaged for July–August of 2010, 2011 and 2012, at altitudes of 15, 16, 17 and 18 km, respectively. It can be seen that the general characteristics of the ASM anticyclone are well represented by EMAC. The anticyclone is very strong in the upper troposphere (at 15 km), and it is still clearly visible in the lower stratosphere (e.g., at 18 km) with the core area shifting northerly with increasing altitude. As found in early studies (e.g., Highwood and Hoskins, 1998), the ASM tropopause is relatively high, with a maximum around 17 km (corresponding to about 92-95 hPa) over the area of approximately 30-120

30   °E and 20-35 °N, including the Iranian Plateau to southern Tibetan Plateau. The air masses in the ASM anticyclone are characterized more by tropospheric than stratospheric properties. This feature has been well reproduced by EMAC using O$_3$ as a stratospheric tracer and CO as a tropospheric tracer, as also done in other model studies (e.g., Gettelman et al.,

2004;Park et al., 2007;Park et al., 2009;Barret et al., 2016;Santee et al., 2017). It was shown recently that EMAC realistically simulates reactive gases and radicals within the ASM anticyclone by comparing the simulation results with aircraft measurements (Lelieveld et al., 2018).

As a general overview, we show the simulated global distributions of the tropospheric burden of various aerosol
components, including BC, OC, dust, sea spray, sulfate, nitrate, ammonium and ALW for July–August 2012 in Fig. 3 (the same global distributions of these aerosols are presented in Figs. S1 and S2 for the years 2010 and 2011). The geographic distribution patterns of the column integrated aerosol properties are similar to those given in the work of Pringle et al. (2010), who also used EMAC but with a different grid resolution (T42L19) and for an earlier simulation period (2001–2002). However, the concentration levels of anthropogenic aerosols, especially sulfate, over the Middle East, India and northeast
China, are significantly higher in our simulations due to increasing emission trends over the period of 2001 to 2010 (Granier et al., 2011). The concentration levels of dust aerosols from our simulation are similar to those of Pringle et al. (2010), but there are some differences in the fine structure, probably due to different meteorological conditions for different periods and/or an improved scheme used in our simulation (Klingmüller et al., 2018).

## 3.2 Aerosol extinction

In situ measurements of aerosols in the UTLS over the Tibetan Plateau have been very sparse. Tobo et al. (2007a, b) firstly observed relatively high number concentrations of sub-micron size aerosols (with radii of 0.15–0.6 μm) near the tropopause region (between about 130 and 70 hPa) during the ASM period with a balloon-borne optical particle counter at Lhasa, China in 1999. He et al. (2014) found a 3–4 km thick aerosol layer in the UTLS peaking at an altitude of 18–19 km (1–2 km above the tropopause), by measuring the aerosol extinction vertical profiles using a micro-pulse lidar at Naqu, China in July–
August 2011. Recently, Yu et al. (2017) further revealed a widespread enhanced aerosol layer extending from several kilometres below the tropopause up to 2 km above the tropopause within the ASM anticyclone, by measuring the vertical profiles of particle surface area density using an optical particle spectrometer in balloon soundings from Kunming, China, in August 2015. By balloon-borne measurements, Brunamonti et al. (2018) found the maximum aerosol backscatter occurring at the cold-point tropopause, revealing the thermodynamically significant levels of the ATAL. Figure 4a shows EMAC
simulated temporal variation in the vertical profile of aerosol extinction over Naqu in August 2011, which coincided with the period of aerosol vertical profile measurements at Naqu reported by He et al. (2014). It can be seen that EMAC predicts persistently enhanced aerosol abundance between 17–20 km altitude, with maxima occurring at 18–19 km; aerosols near the tropopause have a higher variability in the upper troposphere than in the lower stratosphere. A comparison of the model simulated monthly mean aerosol vertical profile with lidar measurements of He et al. (2014) is presented in Fig. 4b. The
aerosol vertical profiles from measurements are shown to have a high variability, especially at high altitudes (i.e., 18–19 km), and simulated aerosol extinction coefficients tend to be at the lower end of the measured range at 18 km altitude and downward.

The ATAL was found by detecting the aerosol optical properties in the UTLS during the ASM (Vernier et al., 2011;Thomason and Vernier, 2013). In this study we also investigate the geographic distributions of model simulated aerosol optical properties together with the associated chemical composition. Figure 5 shows EMAC simulated aerosol extinction coefficients ($K_e$) at altitudes of 15, 16, 17 and 18 km, averaged for July–August of 2010, 2011 and 2012. Unexpectedly, the ATAL, characterized by extensively enhanced aerosol concentrations within the ASM anticyclone as shown by Vernier et al. (2011) and Thomason and Vernier (2013), was not fully reproduced by our simulation. Only over the Tibetan Plateau can the maxima of $K_e$ in the horizontal direction be found, most clearly at 15–16 km altitude (in the upper troposphere) and still visible at 17 km (in the lower stratosphere) in 2010 and 2012. Although the absolute $K_e$ values at 15 km altitude over the Tibetan Plateau are at the same levels in 2011 as those in 2010 and 2012, the enhancement of aerosols is partly shielded by higher $K_e$ values outside the anticyclone (especially at higher latitudes) in 2011. Aerosols outside the anticyclone increased dramatically from the upper troposphere to the lower stratosphere, and as a result even a relative minimum of $K_e$ can be seen within the ASM anticyclone at 17 km in 2011. Previous studies have shown that the Nabro volcano eruption in 2011 has a large impact on stratospheric aerosols over East Asia and the Northern Hemisphere (Sawamura et al., 2012;Uchino et al., 2012). The high $K_e$ values outside the anticyclone in 2011 can be attributed to the effect of the Nabro volcano eruption, which has been taken into account in our model simulation (see Fig. 1 and Sect. 2 above). Note that this "anti-ATAL" phenomenon was also detected by the CALIPSO satellite for July-August 2009 (see Fig. 1d in the work of Vernier et al. (2011)), when a few months earlier the eruption of Sarychev volcano had injected a large plume of $SO_2$ above the tropopause (Vernier et al., 2011).

Mineral dust, WASO and ALW compounds are the dominant aerosol components contributing to aerosol extinction, and BC, OC, and sea spray have very small (1–2 orders of magnitude lower) $K_e$ values in the UTLS within the ASM anticyclone (see Figs. S3, S4 and S5). Figures S6, 6, and 7 show the percent contributions of mineral dust, WASO and ALW to the total $K_e$ values at altitudes of 15, 16, 17 and 18 km in July–August of 2010, 2011 and 2012. The contributions from BC, OC, and sea spray are not shown due to their small values (less than 5% over the Tibetan Plateau). By comparing the chemical aerosol patterns with those of the total $K_e$ shown in Fig. 5, we can see that the broad maxima of aerosol extinction at 15–16 km over the Tibetan Plateau are caused firstly by mineral dust, which contributes approximately 30–60%, and secondly by WASO and associated ALW, which contribute comparably with about 10–40%. There are two maxima in the absolute contribution of WASO to the total $K_e$ in the upper troposphere within the anticyclone in both 2010 and 2012: the first occurring in the eastern part of the anticyclone over the (eastern) Tibetan Plateau, companied by a maximum in the ALW part of $K_e$, and the second in the western part over the Iranian Plateau with no maximum in the extinction by ALW.

High aerosol extinction outside the ASM anticyclone was caused by both WASO and ALW, with WASO making a predominant contribution (typically 50–70%) to the north of the anticyclone and ALW contributing dominantly (up to 60–80%) to the south of it. In contrast to the absolute $K_e$ from mineral dust, which is at the same level in the three years, the absolute $K_e$ contributed by WASO as well as ALW is much higher in 2011 than in 2010 and 2012, especially in the lower stratosphere (e.g., at 17 km), leading to the "anti-ATAL" phenomenon as pointed out above. The pronounced Nabro eruption

of $SO_2$ in 2011 can enhance sulfuric acid and sulfate aerosols in the UTLS, and increase the contributions of WASO and ALW to the aerosol in the UTLS, especially in the lower stratosphere and outside the ASM cyclone, as shown and discussed in the following section.

### 3.3 Ionic aerosols in equivalent

As described in Sect. 2, ionic aerosol composition, including $Na^+$, $K^+$, $Ca^{2+}$, $Mg^{2+}$, $NH_4^+$, $SO_4^{2-}$, $HSO_4^-$, $NO_3^-$, and $Cl^-$ in the liquid phase and their salts in the solid phase, are calculated by the ISORROPIA-II thermodynamic equilibrium model implemented in EMAC. Investigation of these ionic species is helpful for understanding the sources and formation pathways of the aerosols. Figures S7, 8, and 9 present EMAC simulated water soluble ions in the accumulation mode at altitudes of 15, 16, 17 and 18 km for July–August of 2010, 2011 and 2012. Note that (1) the accumulation mode is considered here since nearly all extinction (>99%) is caused by aerosols in this mode; (2) we use "equivalent" (eq) rather than "mole" (mol) or "gram" (g), to express the amount of ions, considering that it may be used more easily to explore the possible formation of a chemical compound from different cations and anions (e.g., 1 mol $NO_3^- = 1$ eq $NO_3^-$, and 1 mol $SO_4^{2-} = 2$ eq $SO_4^{2-}$; (3) $Cl^-$ is not shown here due to its low concentrations (e.g, two orders of magnitude smaller than that of $NO_3^-$) in the investigated regions; (4) for simplicity, $Na^+$, $K^+$ and $Mg^{2+}$ together with $Ca^{2+}$ in the mineral dust are accounted for as $Ca^{2+,*}$ in the equivalent amounts since $Ca^{2+}$ is a typical ionic tracer for mineral dust (Ma et al., 2003). Note that these four ionic species are also contained in sea salt and generally $Na^+$ is taken as a tracer for sea salt. However, mineral dust appears to be a dominant source for all these species in the UTLS over the Tibetan Plateau since they are well correlated with the dust plume and the concentrations of $Ca^{2+}$ are much higher than those of $Na^+$ there.

As shown in Figs. S7, 8, and 9, the levels of $NO_3^-$, $NH_4^+$, and $Ca^{2+,*}$ are enhanced at 15–16 km altitude (in the upper troposphere) within the ASM anticyclone in all three years, and the enhancements are still visible at 17 km (in the lower stratosphere) in 2012. Similar to the WASO contributed $K_e$ in 2010 and 2012 (Figs. S3 and S5), there are two maxima of $NO_3^-$ and $NH_4^+$, one in the eastern part of the anticyclone (over the eastern Tibetan Plateau), and another in the western part (over the Iranian Plateau and the surrounding area). A maximum of $Ca^{2+,*}$ is found over the Tibetan Plateau, coinciding with that of the dust contributed $K_e$ shown in Figs. S3–S5. In contrast, $SO_4^{2-}$, $HSO_4^-$, and $H^+$ are found to be higher outside the anticyclone than inside, mainly due to increased stratospheric sulfate aerosols by the volcanic eruptions, especially in 2011 when the Nabro eruption occurred (Fairlie et al., 2014). The ASM anticyclone not only traps tropospheric pollutants inside but also to some extent blocks intrusions of stratospheric ozone and aerosols from outside. It should be noted that here we only investigate a seasonal mean aspect of the ASM anticyclone. Due to the dynamical instabilities of the ASM anticyclone, entrainment of stratospheric tracers does occur frequently, approximately twice a month (Gottschaldt et al., 2018).

The maximum concentrations of $NH_4^+$ and $NO_3^-$ are shown to occur in the same areas and at the same levels, indicating the existence of $NH_4NO_3$ in the upper troposphere (at 16 and 17 km) and even in the stratosphere (e.g., at 17 km in 2012) within the ASM anticyclone. The areas of high $NH_4^+$ are shown to be wider spread than those of high $NO_3^-$ within the anticyclone, suggesting that some $NH_4^+$ ions are in the form of $(NH_4)_2SO_4$ and partly in $NH_4HSO_4$ (e.g., at 17 km in 2011) in

the corresponding areas. The levels of nitrate ($NO_3^-$) and sulfate ($SO_4^{2-}$) within the anticyclone can be either similar or rather different from each other, depending on the altitude and simulation year. For example, higher $NO_3^-$ than $SO_4^{2-}$ is simulated at 15 km altitude within the anticyclone, in particular over the western area of the Iranian Plateau; at 17 km, the simulated $NO_3^-$ concentrations within the anticyclone are still higher in 2012, but are much smaller than those of $SO_4^{2-}$ in 2010 and 2011.

Due to the effect of the Nabro eruption, higher $SO_4^{2-}$, $HSO_4^-$ and $H^+$ concentrations are found in the year 2011 than in 2010 and 2012 over the ASM region and the whole northern hemisphere, most profoundly in the lower stratosphere. The distribution patterns of $SO_4^{2-}$ and $HSO_4^-$ indicate that considerable fractions of sulfate aerosols can enter into the anticyclone from outside, mainly through the northern and eastern edges; these sulfate aerosols might be quite acidic in the lower stratosphere, as indicated by high $H^+$ concentrations, resulting in a deficiency of $NO_3^-$ in the aerosol phase within the

anticyclone (e.g., at 17 km in 2011).

In addition to ammonia, the reaction with mineral dust can also provide a pathway for nitrate formation due to neutralization by the crustal cations such as $Ca^{2+}$ (Ma et al., 2003). An investigation of the fine distribution structures of $NO_3^-$, $NH_4^+$, and $Ca^{2+,*}$ over the Tibetan Plateau shows that large residuals of $NO_3^-$ minus $NH_4^+$ exist in some areas (e.g., at 15–16 km in 2010 and 2012), where the maxima of $Ca^{2+,*}$ occur as well. This finding indicates that multiphase reactions of

gaseous nitric acid ($HNO_3$) have taken place on the surface of mineral dust, leading to the formation of $Ca(NO_3)_2$ or analogs in the upper troposphere over the Tibetan Plateau. Considerable amounts of $Ca^{2+,*}$ are also found in the lower stratosphere (e.g., at 17 km in 2010 and 2011) within the ASM anticyclone; however, these $Ca^{2+,*}$ cations should exist in the form of $CaSO_4$ (gypsum) or analogs since $NO_3^-$ is deficient.

As demonstrated in previous studies, lightning $NO_x$ clearly dominates the $NO_x$ budget from the tropopause to 100 hPa

below it, and its emissions are much stronger in the Tibetan part (Gottschaldt et al., 2018). Lightning $NO_x$ also plays a central role in sustaining upper tropospheric OH concentrations over the monsoon (Lelieveld et al., 2018). In our simulations, lightning $NO_x$ emissions within the ASM anticyclone (20-140 °E and 10-45 °N) are very intensive, with estimated values of 1,1, 0.5 and 0.9 Tg (N) $a^{-1}$ in the years 2010, 2011 and 2012, respectively. High $NO_x$ and OH concentrations are in favour of $HNO_3$ formation via the reaction $NO_2$ + OH, which may provide an import source of nitrate within the anticyclone. Tost

(2017) found 60% of aerosol nitrate between 500 hPa and the tropopause being produced from lightning in the ASM and its outflow under the present climatic condition. In addition to the available nitric acid, other factors such as neutralising ions (e.g., $Ca^{2+}$, $NH_4^+$ and $SO_4^{2-}$) and temperature can also influence the amount of nitrate aerosols in the ATAL. How these factors and associated processes affect nitrate aerosols in the UTLS over the Tibetan Plateau will be investigated thoroughly in a future study.

**3.4 Aerosol mass concentrations**

Figures S8, 10 and 11 present EMAC simulated mass concentrations of aerosols in the accumulation mode, including specifically BC, OC, mineral dust, sulfate, nitrate, ammonium, and ALW at altitudes of 15, 16, 17 and 18 km, averaged for July–August 2010, 2011 and 2012. For simplicity, the crustal species $Ca^{2+}$, $Na^+$, $K^+$ and $Mg^{2+}$ are not shown individually,

considering that these ionic species within the ASM anticyclone originate predominantly from mineral dust, having the same constant relative fractions as their emission sources do (see Sect. 2 above). Sulfate encompasses both $SO_4^{2-}$ and $HSO_4^-$, the latter at relatively low concentrations, and is thus expressed as $SO_4^{2-}$ as well. Other species such as sea spray components are not shown due to their extremely low mass concentrations in the UTLS over the Tibetan Plateau. The geographical

distribution features of major aerosols in the UTLS over the Tibetan and within the ASM anticyclone have been presented by their extinction or ionic species in Sects. 3.2 and 3.3. Here we use aerosol mass concentrations for two purposes: (1) to further quantitatively evaluate the relative importance of different aerosol components over the Tibetan Plateau; (2) to make a comparison of EMAC simulated major chemical components with previous studies, where mass concentrations were widely used.

It is shown that mineral dust, sulfate and nitrate are three major components of dry aerosols in the UTLS within the ASM anticyclone. Ammonium is not mentioned here considering that it can exist in the form of sulfate or nitrate, as discussed in Sect. 3.3. Over the Tibetan Plateau mineral dust is simulated to be a dominant component in the UTLS, except for the lower stratosphere in 2011 when the Nabro eruption increased sulfate aerosols dramatically, which is in agreement with measurements by He et al. (2014). Li et al. (2005) predicted an enhancement of aerosols within the ASM anticyclone using

GEOS-Chem, which considered only BC, OC and sulfate aerosols. Neely et al. (2014) only considered sulfate aerosols in their model study of the ATAL using WACCM3. They demonstrated that if moderate volcanic activity was included in WACCM3, aerosol extinction from sulfate could be simulated to be higher outside than inside the anticyclone (see Fig. 5 of Neely et al. (2014) ), in good agreement with the simulation results from this study.

Fadnavis et al. (2013) simulated aerosols in the UTLS during the ASM using ECHAM5-HAMMOZ (with the resolution

T42 corresponding to 2.75°×2.75° in the horizontal dimension), into which M7 (the same as in the present study) had been implemented to describe the size distribution of aerosols, including BC, OC, sulfate, sea salt and mineral dust, but without considering nitrate (which has been included in EMAC and used in the present study). Of all aerosol types around the tropopause within the ASM anticyclone, mineral dust was simulated to occur at highest concentrations, followed by sulfate (see Fig. 1 of Fadnavis et al. (2013)). This result is in fair agreement with the order of major components derived in this

study, although nitrate was not considered in the work of Fadnavis et al. (2013). Fadnavis et al. (2013) predicted a maximum seasonal mean dust aerosol concentration of ~30 ng m$^{-3}$ at 110 hPa, much lower than the value (>100 ng m$^{-3}$) at 16 km over the Tibetan Plateau as simulated in this study (Figs. S8, 10 and 11), possibly related not only to the fact that the updated EMAC dust emission scheme takes the topography into account, which has been shown to enhance the emissions from basins and valleys, in better agreement with observations (Klingmüller et al., 2018), but also to the higher resolution used in

this study. The geographic distribution pattern of sulfate from our simulation is quite different from that of Fadnavis et al. (2013) since volcanic sulfate was most likely not considered in their simulation.

Yu et al. (2015) explored the composition and optical properties of the ATAL using CESM1 (with 2° horizontal resolution) coupled with a sectional aerosol microphysics model including primary emitted organics, secondary organics, dust, sea salt, black carbon, and sulfate. Yu et al. (2015) predicted an aerosol enhancement mainly composed of mineral dust

extending from the surface up to 13 km above Africa. However, although mineral dust was included in their simulation, Yu et al. (2015) suggested the ATAL (between 100–230 hPa) to be mostly composed of organics (roughly 60%) and sulfate (roughly 40%) by mass given 2010 emissions (See Fig. 3 of Yu et al. (2015)). Our simulation results show that OC has a much smaller contribution to the ATAL than either dust or sulfate (or nitrate), in contrast to Yu et al. (2015). Similar to the comparison with Fadnavis et al. (2013) mentioned above, the different dust emission scheme and model resolution used by Yu et al. (2015) might lead to significant discrepancy of dust concentration in the UTLS within the ASM anticyclone compared to this study. Moreover, Yu et al. (2015) did not take into account stratospheric volcanic emissions and nitrate aerosols in their simulation, which might underestimate the contribution of inorganic aerosols to the ATAL with respect to this study.

Gu et al. (2016) performed model simulations using GEOS-Chem (with 2° latitude by 2.5° longitude horizontal resolution), which considers detailed tropospheric chemistry coupled with aerosols including sulfate, nitrate, ammonium, BC, OC, sea salt and mineral dust. They showed elevated sulfate, nitrate, ammonium, BC and OC in the UTLS over the ASM region in summer and concluded that nitrate is the most dominant aerosol composition in the UTLS over the studied region. In the work of Gu et al. (2016), concentrations of mineral dust were simulated to be 5.0–7.0 ng m$^{-3}$ over the studied region and contribute less than 5.0% to total aerosol mass at 100 hPa. It turns out that there is a large discrepancy between Gu et al. (2016) and this study regarding the contribution of mineral dust to the enhancement of aerosols over the Tibetan Plateau and ASM region. Similar to Gu et al. (2016), we also simulate higher nitrate at 16–18 km within the ASM anticyclone in one of the simulated years, i.e. 2012, after March when volcanos are considered as quiescent arbitrarily in our simulation due to limited MIPAS data. However, the distribution patterns of nitrate in the UTLS are rather different between the two studies. Firstly, we found two nitrate maxima, located over the eastern and western parts of the anticyclone, respectively, which was not simulated (or shown clearly) by Gu et al. (2016). Secondly, an elongated, very high nitrate belt at the northern edge of the anticyclone around 40°N, as shown in Fig. 7c of Gu et al. (2016), is not reproduced by our simulation. Note that persistent and widespread more acidic conditions outside the anticyclone, as simulated in this study by including the volcanic SO$_2$ eruption effects, do not favor the presence of ammonium nitrate that was reported by Gu et al. (2016).

## 3.5 Transport of dust, ammonia and water vapor

According to our EMAC simulations, mineral dust aerosols in the accumulation mode make a major contribution to the total aerosol extinction in the UTLS with the ASM anticyclone, contributing predominantly to the maximum $K_e$ at 15–16 km over the Tibetan Plateau. Here we investigate the spatial distribution of these dust aerosols to understand from where they originate. Figure 12 presents EMAC simulated dust mass concentrations in the accumulation mode at different altitudes from the surface to 18 km, averaged for July–August 2012. Although there are some small differences in the fine structure of the mass concentration distribution, the seasonally averaged transport patterns of dust over the Tibetan Plateau and the surrounding area are the same in 2010 and 2011 (see Figs. S9 and S10) as in 2012. The geographical distribution of mineral dust at the surface shows that the emission sources are widely spread, e.g. from northern Africa, the Arabian Peninsula and

northern India, to western China and Mongolia. There is a broad maximum of dust surface concentration at the northern edge of the Tibetan Plateau, located at the northern slope of the middle range (77–86 °E) of the Kunlun Mountains running from west to east, to the north at the Tarim Basin. The enhanced dust aerosol concentrations from different source areas are clearly visible at 6 km, and their regional impacts can still be seen at 8 km altitude. At 10 km altitude, the enhancement of dust

aerosols over the Tibetan Plateau are still remarkable, with a lofted dust plume from the local emission source in the northwest of the Tibetan Plateau. At altitudes above 10 km, up to 16 km, enhanced dust aerosols persist over the Tibetan Plateau, with the maximum shifting eastward and then southward under the influence of anticyclonic circulation. The plume transport pattern that connects the maximums of mineral dust aerosols over the Tibetan Plateau to the surface indicates that dust aerosols in the UTLS over the Tibetan Plateau are predominantly from emission sources at the northern Tibetan Plateau

and its northern slope areas.

The distribution pattern of dust near the surface on the Tibetan Plateau and the nearby areas shown above (e.g., in Fig. 12) is generally similar to that of the tropospheric column concentration (e.g., Fig. 3). However, the position of maximum values for the latter shifts slightly to the north, where larger amounts of dust in the total column may be attributed to the Taklamakan Desert in the Tarim Basin than the central Tibetan Plateau. Satellite data analysis by Xu et al. (2015) has also

shown high aerosol optical depth (AOD) values in the same area to the north of the Tibetan Plateau (see Fig. 2 in the paper of Xu et al. (2015)). Note that higher AOD to the south than the north of the Tibetan Plateau shown by Xu et al. (2015) might be attributed to much greater contributions of anthropogenic aerosols from India; moreover, only clear days are considered for satellite data products, which tends to affect the seasonal statistics more for aerosols in the southern Tibetan Plateau, where there are more clouds and precipitation associated with the ASM (Zhao et al., 2018).

Our simulation results support the arguments of previous studies (Huang et al., 2007;Xia et al., 2008;Xu et al., 2015) in that mineral dust aerosols are the dominant aerosol type on the Tibetan Plateau and they are most likely transported to the main body of the Tibetan Plateau across the northern edge. Here we emphasize that compared to the transport from the interior of the Taklamakan Desert in the Tarim Basin, direct emissions of dust aerosols from local sources in the northern Tibetan Plateau, i.e. at the northern slope (around 4-5 km altitude) of the middle Kunlun Mountains, are more likely to

contribute to the maximum. The dust layers detected by satellite have been reported to extend up to 8–10 km altitude over the northern Tibetan Plateau in summer (Huang et al., 2007;Liu et al., 2008;Xu et al., 2015). Our simulation shows that the maximum of dust at 10 km altitude is still located just over the strongest source area at the surface, agreeing with satellite measurement results and indicating that dust aerosols are lofted directly to the upper troposphere along the northern slope of the Tibetan Plateau. At 10 km altitude and above, the position of maximum dust aerosols travels along the anticyclonic

circulation, mainly within its core area.

Brühl et al. (2018) showed high sensitivity of mineral dust reaching the UTLS to model resolution, owing mostly to the differences in convection top height and overshooting convection in the parametrizations. Compared to T63L90 (with 1.88° horizontal resolutionand 90 vertical layers), which fits best to the observations (including the profile shown in Fig. 4) as for which the convection parameterization was developed (Tiedtke, 1989), EMAC tends to overestimate stratospheric DAOD

with simulation at lower resolution (e.g., T42L90 ), and underestimate it at higher resolution (e.g., T106L90) (Brühl et al., 2018). We find that the aerosol extinction levels in the UTLS within the ASM anticyclone of this study are systematically underestimated with respect to the results of Brühl et al. (2018) with T63L90. For example, the maximum monthly mean $K_e$ at 16 km over the Tibetan Plateau from the T63L90 simulation is about 3 times larger than that from the T106L90 simulation. Such difference might partly be attributed to the difference in simulated mineral dust, but also it is most likely due to less efficiently convective transport of anthropogenic aerosols and their gaseous precursors in the T106L90 simulations compared to T63L90 simulations. It should be noted that deep convection events occur much less frequently over the northern part of the Tibetan Plateau than the southern part of it, to the latter pollution from South Asia tends to accumulate (Lelieveld et al., 2018). Figures S11, S12 and 13 present the geographical distributions of specific humidity, $q_v$, at various altitudes from the surface to 18 km over the ASM region from our EMAC simulations, averaged for July–August in the years 2010, 2011 and 2013, respectively. It is shown that deep convection of moisture occurs over the southern edge of the Tibetan Plateau, with the maximum $q_v$ attained from 10–11 km altitude upward. This is similar to the reanalysis results from previous studies, e.g., Xu et al. (2014), who found the most significantly relative enhanced $q_v$ at 250–300 hPa over the same area. The geographical position of the mineral dust maximum in the upper troposphere over the Tibetan Plateau is apparently different from that of the moisture maximum, indicating that different pathways for the transport of aerosols and their gaseous precursors from the lower troposphere to the UTLS over the Tibetan Plateau.

Figures S13, S14 and 14 present the geographical distributions of gaseous ammonia (NH$_3$) at various altitudes from the surface to 18 km over the ASM region from our EMAC simulations, averaged for July–August in the years 2010, 2011 and 2013, respectively. In addition to northern India, which is located to the southwest of the Tibetan Plateau, an enhancement of surface NH$_3$ is also found in the central main body of the Tibetan Plateau, although the maximum NH$_3$ mixing ratio in the latter area (~10 ppbv) is about half of that in the former area (~20 ppbv). It is shown that surface ammonia can be more efficiently convected and transportd to the UTLS from the Tibetan Plateau than from the Indian region, leading to high levels of NH$_3$, e.g., about 0.1 ppbv at 17 km altitude. Such large amounts of lofted ammonia accumulate within the ASM anticyclone, providing the basic gas favourable for the formation of nitrate aerosols (especially in the years 2010 and 2012 when the volcanic eruption effect is relatively small with respect to 2011), as discussed in Sect. 3.3. Our EMAC simulation at a relatively high resolution (i.e., T106L90) reveals clearly the important role of emissions and the orographic forcing and deep convection in transport of mineral dust and gaseous precursors like NH$_3$ over the Tibetan Plateau. T106L90 with improved convection parameterization is suggested to investigate the transport of aerosols and their gaseous precursors associated with complex topography and finer structure of the anticyclone.

## 4 Conclusions

We have investigated the chemical components and their sources of aerosols in the UTLS over the Tibetan Plateau during summer, within the ASM anticyclone, using the atmospheric chemistry general circulation model EMAC at T106L90 resolution. The model simulation performed for the period from January 2010 to December 2012, and for this study the

seasonally (July–August) averaged results are analyzed, focusing on the spatial distribution characteristics of aerosols extinction and chemistry at different altitudes around the tropopause. Model evaluation against lidar measurements over the central Tibetan Plateau shows that EMAC can reproduce the basic vertical distribution feature of summertime aerosols in the UTLS over the Tibetan, although simulated aerosol extinction coefficients tend to be at the lower end of the measured range.

5    Our simulation results show that over the Tibetan Plateau, there is a maximum in aerosol extinction in the upper troposphere for all three investigated years (at 15–16 km in 2010–2012), and even in the lower stratosphere (at 17 km in 2010 and 2012) except for 2011, when the Nabro eruption occurred (in June). It is found that the maximum aerosol extinction over the Tibetan Plateau can be attributed mainly to mineral dust aerosols (with a contribution of 30-60% at 15–16 km), followed by water soluble aerosols and associated liquid water (each making a comparable contribution of 10–40% at 10    15–16 km), and these three components also dominate aerosol extinction in the UTLS within the ASM anticyclone. In contrast to the absolute $K_e$ from mineral dust, which is at the same level in the three years, the absolute $K_e$ contributed by water-soluble species (WASO) and aerosol liquid water (ALW) is much higher in 2011 than in 2010 and 2012, especially in the lower stratosphere (e.g., at 17 km). The enhancement of aerosols around the tropopause within the ASM anticyclone in 2011 is less pronounced, due to shielding by high background sulfate aerosols outside the anticyclone. We even find an 15    "anti-ATAL" phenomenon characterized by a relative minimum of aerosol extinction in the lower stratosphere (e.g., at 17 km in 2011) due to the strong volcano effect. This indicates that the ASM anticyclone not only efficiently traps tropospheric pollutants inside but also blocks the intrusions of stratospheric ozone and aerosols from outside.

It is shown that mineral dust, sulfate and nitrate are three major components of dry aerosols in the UTLS within the ASM anticyclone. The mineral dust and nitrate concentrations decrease from the upper troposphere to the lower stratosphere, and 20    vice versa for sulfate due to greater volcano influence in the lower stratosphere than in the upper troposphere. The levels of $NO_3^-$, $NH_4^+$, and crustal cations like $Ca^{2+}$ are strongly enhanced in the upper troposphere (at 15–16 km altitudes) within the ASM anticyclone, and the enhancements persist in the lower stratosphere during volcanically quiescent times (e.g., at 17 km in 2012). In contrast, sulfate aerosol ($SO_4^{2-}$ and $HSO_4^-$) concentrations in the lower stratosphere are found to be generally higher outside the anticyclone than inside, especially in 2011, due to volcano eruptions occurring outside the ASM area. We 25    find two maxima of $NO_3^-$ and $NH_4^+$ in the UTLS within the ASM anticyclone, one over the east of the Tibetan Plateau, and another over the Iranian Plateau and its surrounding area. A maximum of crustal cations is found to be located over the Tibetan Plateau, coincided with that of mineral dust. Our ionic aerosol composition analysis reveals the existence of $Ca(NO_3)_2$ or analogs (from $Mg^{2+}$, for example) in the upper troposphere over the Tibetan Plateau due to multiphase reactions on mineral dust, and substantial amounts of $NH_4NO_3$ within the ASM anticyclone. In the stratosphere these cations are most 30    likely to exist in the form of $CaSO_4$ (or analogs) and $(NH_4)_2SO_4$ (partly as $NH_4HSO_4$) since $NO_3^-$ is expelled from the aerosol under acidic conditions, especially when the impact from volcano eruptions is large. We find considerable influence of stratospheric aerosols in the anticyclone from its surroundings, mainly through the northern and eastern edges.

We find that dust aerosols in the UTLS within the ASM anticyclone are predominantly from the emission sources in the northern Tibetan Plateau and its northern slope areas. Compared to the interior of the Taklamakan Desert in the Tarim Basin

(located to the north of the Tibetan Plateau), the areas at the northern slope of the middle Kunlun Mountains (located in the northern Tibetan Plateau) are more likely to be a predominant source of dust aerosols over the Tibetan Plateau. Our simulations indicate that dust aerosols accumulated in these source areas can be lofted directly to the upper troposphere along the northern slope of the Tibetan Plateau, up to an altitude of at least 10 km. At 10 km altitude and above, lofted dust aerosols are transported eastward and then southward under the influence of anticyclonic circulation and most of them entrain into the core area of anticyclone. We also find that the emissions and convective transport of ammonia in the central main body of the Tibetan Plateau contribute significantly to the enhancement of gaseous $NH_3$ in the UTLS over the Tibetan Plateau and entire ASM anticyclone area.

Compared with previous model studies, the EMAC model version we used for this study appears to be the most complete in the treatment of aerosol sources, microphysics and chemistry. We find a larger dust contribution to the ATAL after updating the dust emission scheme. Our results are also consistent with the recently reported observation of enhanced nitrate (Höpfner et al, 2019), which appears to be missing in previous studies. Both the ATAL and Anti-ATAL characteristics are well simulated by our model in general. Nevertheless, further model improvements are desirable, e.g., of convection processes, which at T106L90 might underestimate the transport of pollution to the UTLS from the boundary layer to some extent. While recent field campaigns have been carried out to measure aerosols and trace gases in the UTLS in the southern and/or western parts of the ASM anticyclone (Lelieveld et al., 2018;Vernier et al., 2018), intensive measurements over the Tibetan Plateau are still sparse and strongly recommended  (Zhao et al., 2018). This study highlights the important role of the northern Tibetan Plateau in the emissions and transport of dust aerosols to the anticyclone. We argue that the UTLS over the Tibetan Plateau acts as a well-defined conduit, not only for pollutants but also for natural aerosols and gases. Interactions of these natural and anthropogenic aerosols and gases in the UTLS are intricately connected and need further investigations.

**Code and data availability**

The Modular Earth Submodel System (MESSy) is continuously further developed and applied by a consortium of institutions. The usage of MESSy and access to the source code is licensed to all affiliates of institutions, which are members of the MESSy Consortium. Institutions can become a member of the MESSy Consortium by signing the MESSy Memorandum of Understanding. More information can be found on the MESSy Consortium website (http://www.messy-interface.org, last access: 15 August 2019). The ECHAM climate model is available to the scientific community under the MPI-M Software License Agreement (https://www.mpimet.mpg.de/en/science/models/license, last access: 16 July 2019). The simulations results analyzed here are archived at CAMS, Beijing, and available on request.

## Author contributions

JM, XX, XZ, KA, and JL designed the study. AP, CB, VK, KK, and HT provided the EMAC and submodel code. JM performed the model simulations with support from BS, CB and AP. QH provided the aerosol vertical profile data from lidar measurements. BC, YJ, NL, and PY gave assistance in dust source identification and meteorological data analysis. JM

prepared the manuscript with contributions from all co-authors.

## Competing interests

The authors declare that they have no conflict of interest.

## Special issue statement

This article is part of the special issue "Study of ozone, aerosols and radiation over the Tibetan Plateau (SOAR-TP)

(ACP/AMT inter-journal SI)". It is not associated with a conference.

## Acknowledgements

This work was funded by the National Natural Science Foundation of China (grant nos. 91537107, 91837311 and 41875146). We thank the International Scientific Partnership Program (ISPP) of the King Saud University for supporting the research. JM would like to thank Patrick Jöckel, Rolf Sander, Astrid Kerkweg, Sergey Gromov, Hendrik Merx, and other MPIC

and/or MESSy colleagues for their help in using EMAC and the submodels.

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

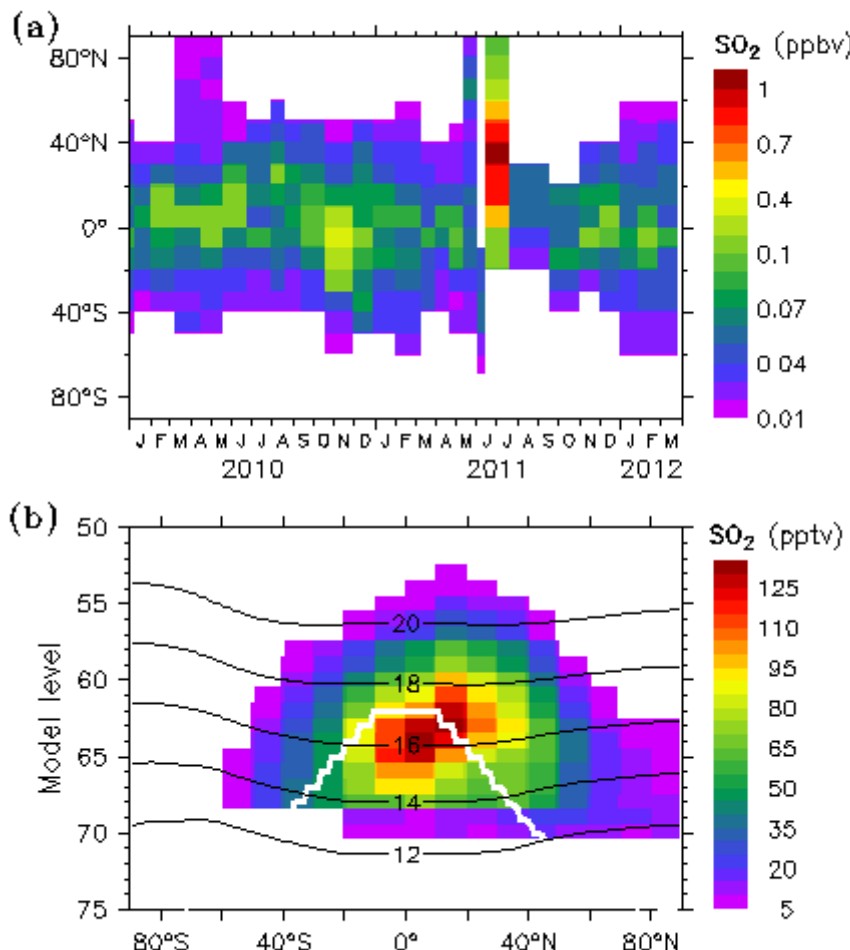

**Figure 1.** MIPAS observed SO$_2$ in the volcano eruption plumes during January 2010 - March 2012, projected onto the EMAC model grid and averaged in the zonal direction for the plot. **(a)** Latitudinal distribution of model level 60–68 (approximately 14–18 km a.s.l.) averaged SO$_2$ (in nmol mol$^{-1}$ or ppbv), as a function of volcano event time. The grid width for each event represents the time interval between the previous and next event. **(b)** Latitudinal and vertical distribution of SO$_2$ (in pmol mol$^{-1}$ or pptv), averaged for all volcano events. Black lines are altitude contours (in km a.s.l.), and the white line highlights the WMO tropopause. The blank refers to the areas where the volcano effects on SO$_2$ were not detected by MIPAS.

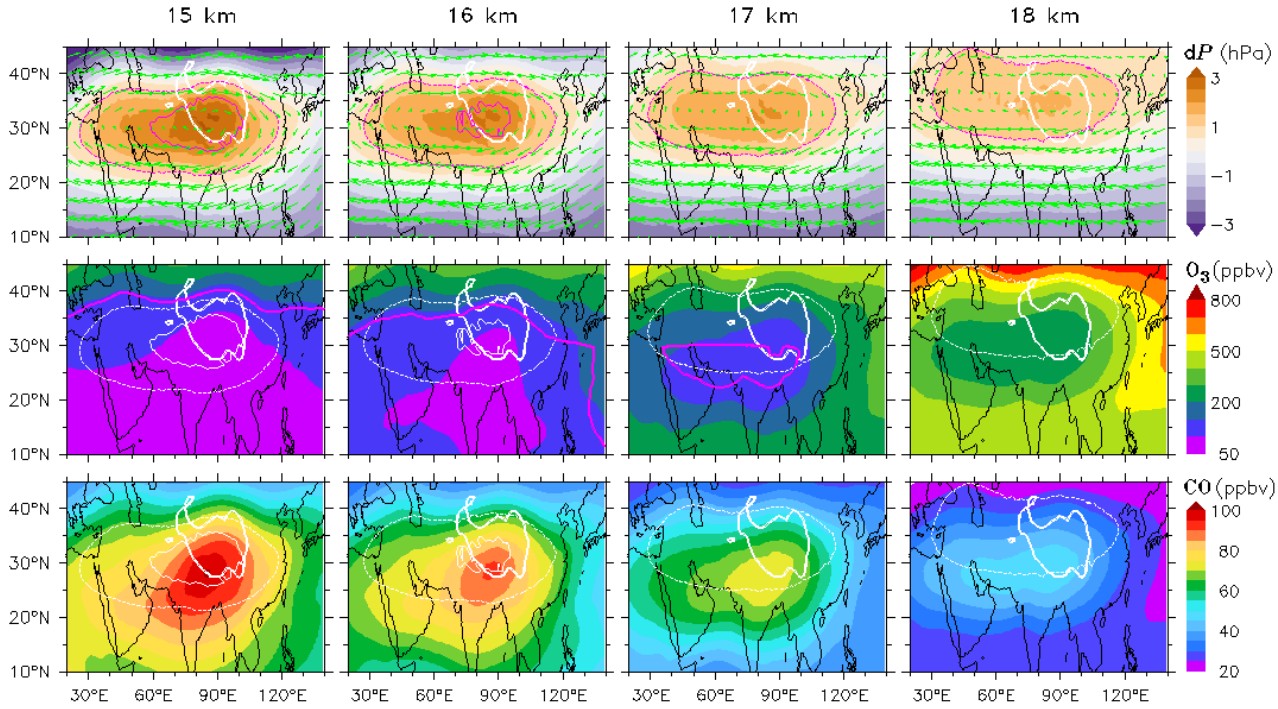

**Figure 2.** Pressure deviation, d$P$, in units of hPa **(top)**, $O_3$ in units of nmol mol$^{-1}$ or ppbv **(middle)** and CO in units of nmol mol$^{-1}$ or ppbv **(bottom)** at altitudes of 15, 16, 17 and 18 km a.s.l. **(first to forth column)**, averaged for July–August of 2010, 2011 and 2012, from EMAC simulations. Pressure deviation, i.e. d$P = P - P_{avg}$, refers to the difference of pressure, $P$, at each model grid relative to the regionally averaged value, $P_{avg}$, at the same altitude, with $P_{avg}$ values of 132.1, 111.4, 93.6 and 78.7 hPa at 15, 16, 17 and 18 km altitudes, respectively. Green arrows indicate the wind field **(top)**, and purple solid lines the position of the WMO tropopause **(middle)**. Purple **(top)** and white **(middle** and **bottom)** thin dashed lines (and also solid lines) are the pressure deviation contour of 1 hPa (and 1.3 hPa), highlighting the anticyclone area (and its core area, applicable to the 15 km panels). Thick white lines refer to the terrain height contour of 3 km, highlighting the Tibetan Plateau area.

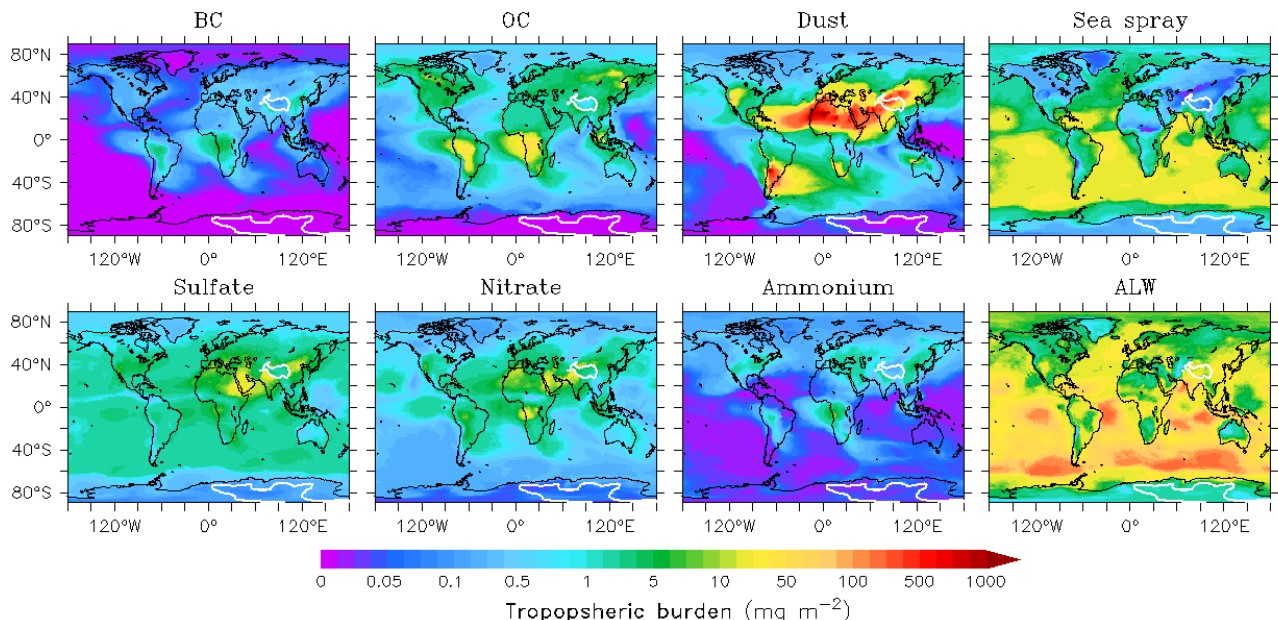

**Figure 3.** EMAC simulated global distributions of the tropospheric burden (in units of mg m$^{-2}$) of black carbon (BC), organic compounds (OC), mineral dust, sea spray, sulfate, nitrate, ammonium and aerosol liquid water (ALW), averaged for July–August 2012. White lines refer to the terrain height contour of 3 km, highlighting the Tibetan Plateau area in the northern hemisphere.

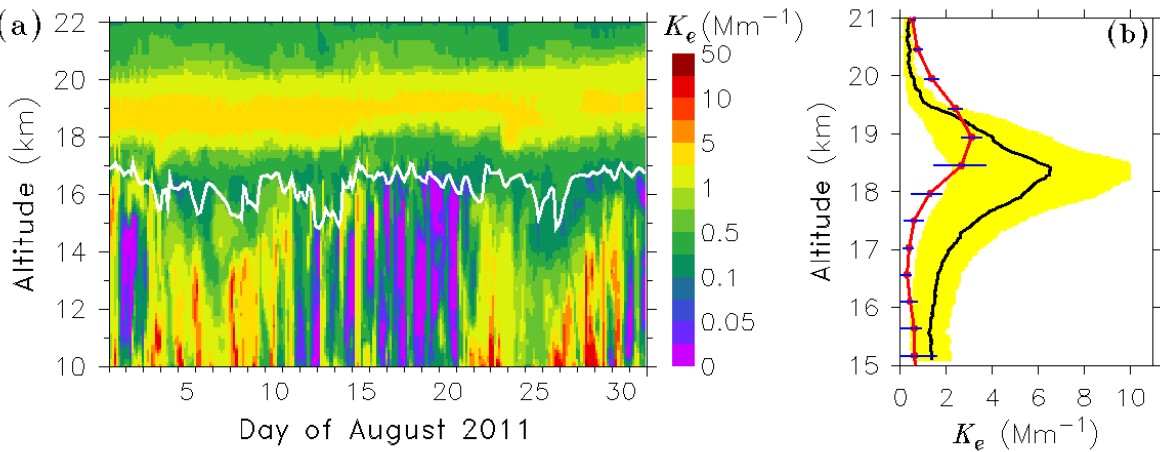

**Figure 4. (a)** Time series of the vertical profile of aerosol extinction coefficient at 550 nm wavelength, $K_e$, in unit of inverse of megameter (Mm$^{-1}$ or 10$^{-6}$ m$^{-1}$) over Naqu at three-hour intervals in August 2011, as simulated by EMAC. Thick white line refers to the WMO tropopause altitude in unit of km a.s.l.. **(b)** Comparison of the model simulated vertical distribution of $K_e$ in the UTLS region with lidar measurements at Naqu. Red and blue lines refer to the monthly mean values and standard deviations derived from the model. Black line and yellow shaded area are the mean values and standard deviation ranges of measurement data.

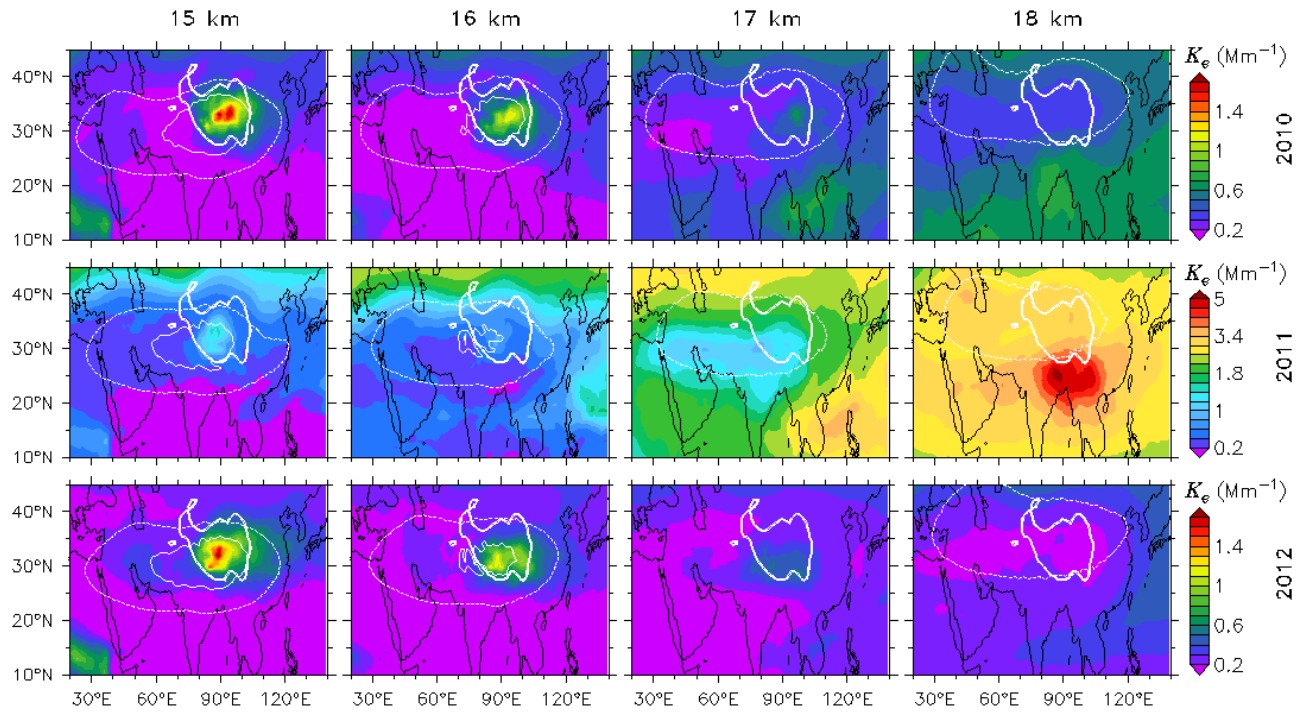

**Figure 5.** EMAC simulated aerosol extinction coefficients at 550 nm wavelength, $K_e$, in units of Mm[-1] (i.e., $10^{-6}$ m[-1]) at altitudes of 15, 16, 17 and 18 km a.s.l. **(first to forth column)**, averaged for July–August of 2010 **(top)**, 2011 **(middle)** and 2012 **(bottom)**. Thin white lines indicate the anticyclone area, with the same index as used in Fig. 2. Thick white lines refer to the terrain height contour of 3 km, highlighting the Tibetan Plateau area. Note that a different colour bar scale is used for $K_e$ in 2011 from that in 2010 and 2012.

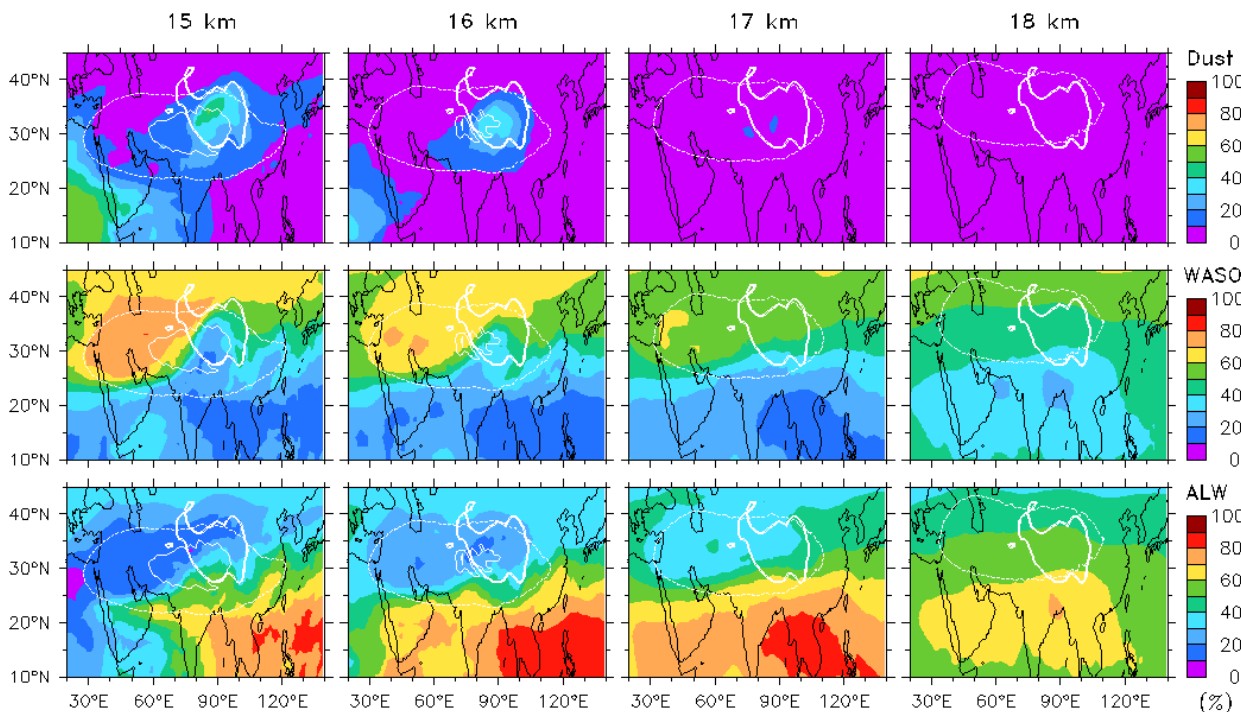

**Figure 6.** Percent contributions of mineral dust **(top)**, water-soluble species (WASO) **(middle)** and aerosol liquid water (ALW) **(bottom)** to the total aerosol extinction coefficient at altitudes of 15, 16, 17 and 18 km a.s.l. **(first to forth column)**, during July–August 2011. Thin white lines indicate the anticyclone area, with the same index as used in Fig. 2, and thick white lines highlight the Tibetan Plateau area.

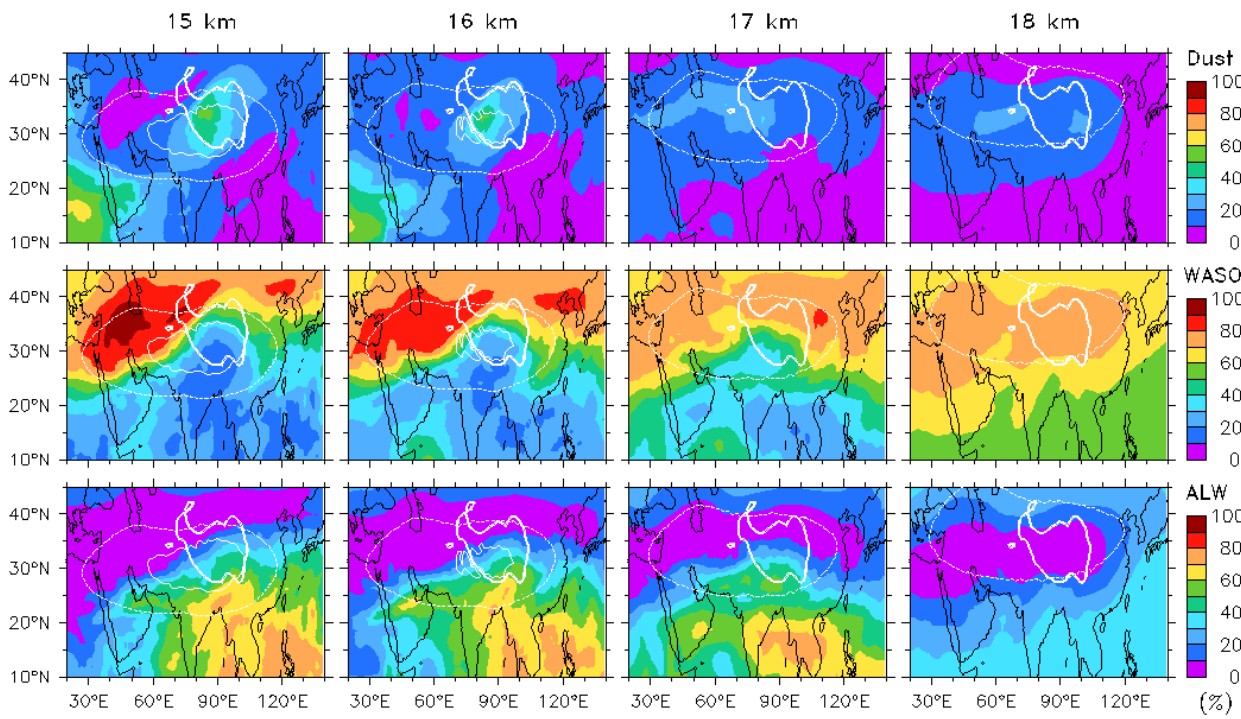

**Figure 7.** Same as Fig. 6, but for the year 2012.

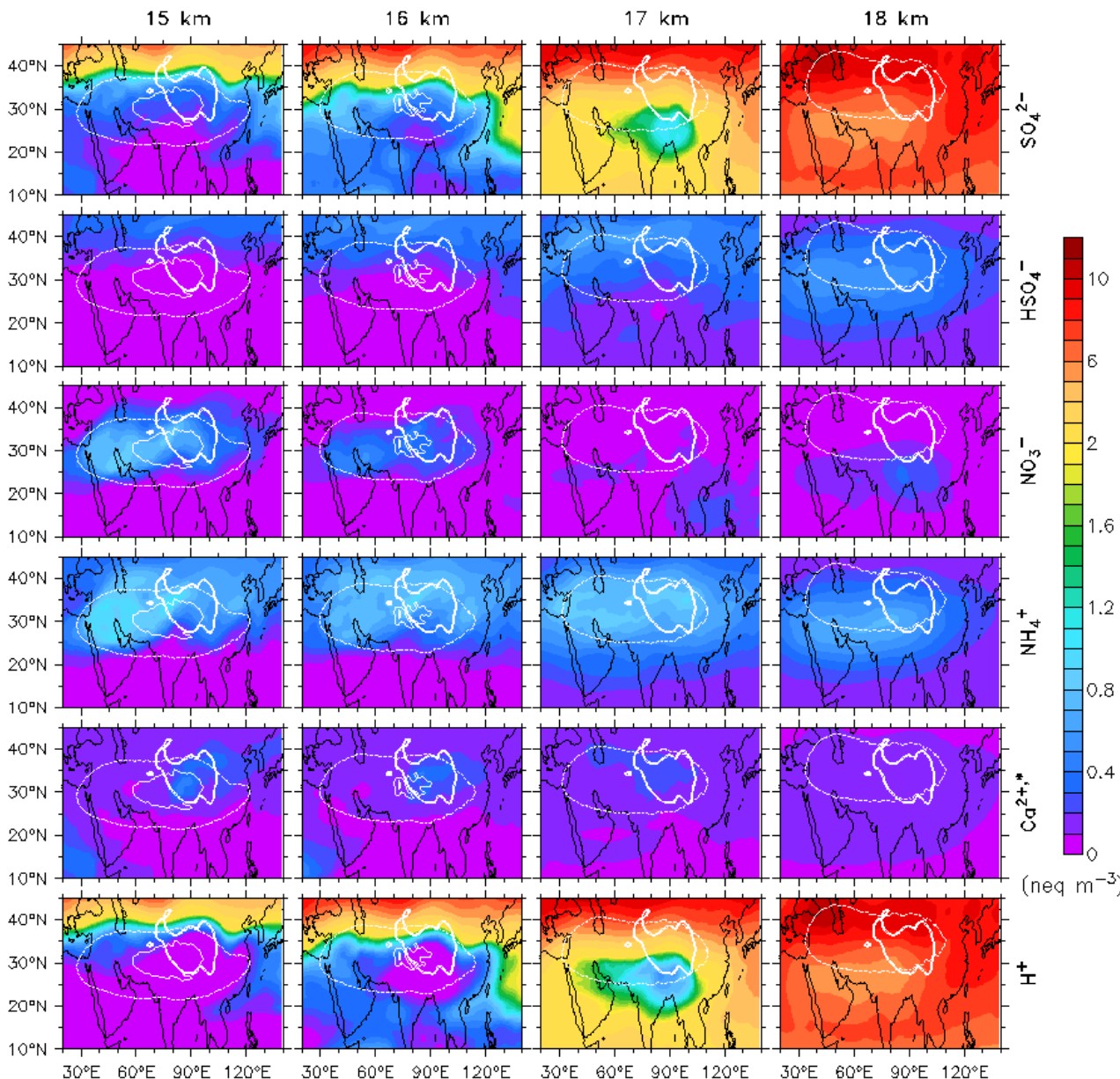

**Figure 8.** EMAC simulated water soluble ionic aerosols in the accumulation mode, including specifically $SO_4^{2-}$, $HSO_4^-$, $NO_3^-$, $NH_4^+$, $Ca^{2+(*)}$, and $H^+$ (**first to sixth row**) in units of nano-equivalent per cubic meter of air (neq m$^{-3}$), at altitudes of 15, 16, 17 and 18 km a.s.l. (**first to forth column**), averaged for July–August 2011. Note that besides $Ca^{2+}$, $Na^+$, $K^+$ and $Mg^{2+}$ are also accounted for in $Ca^{2+(*)}$. Thin white lines indicate the anticyclone area, with the same index as used in Fig. 2, and thick white lines highlight the Tibetan Plateau area.

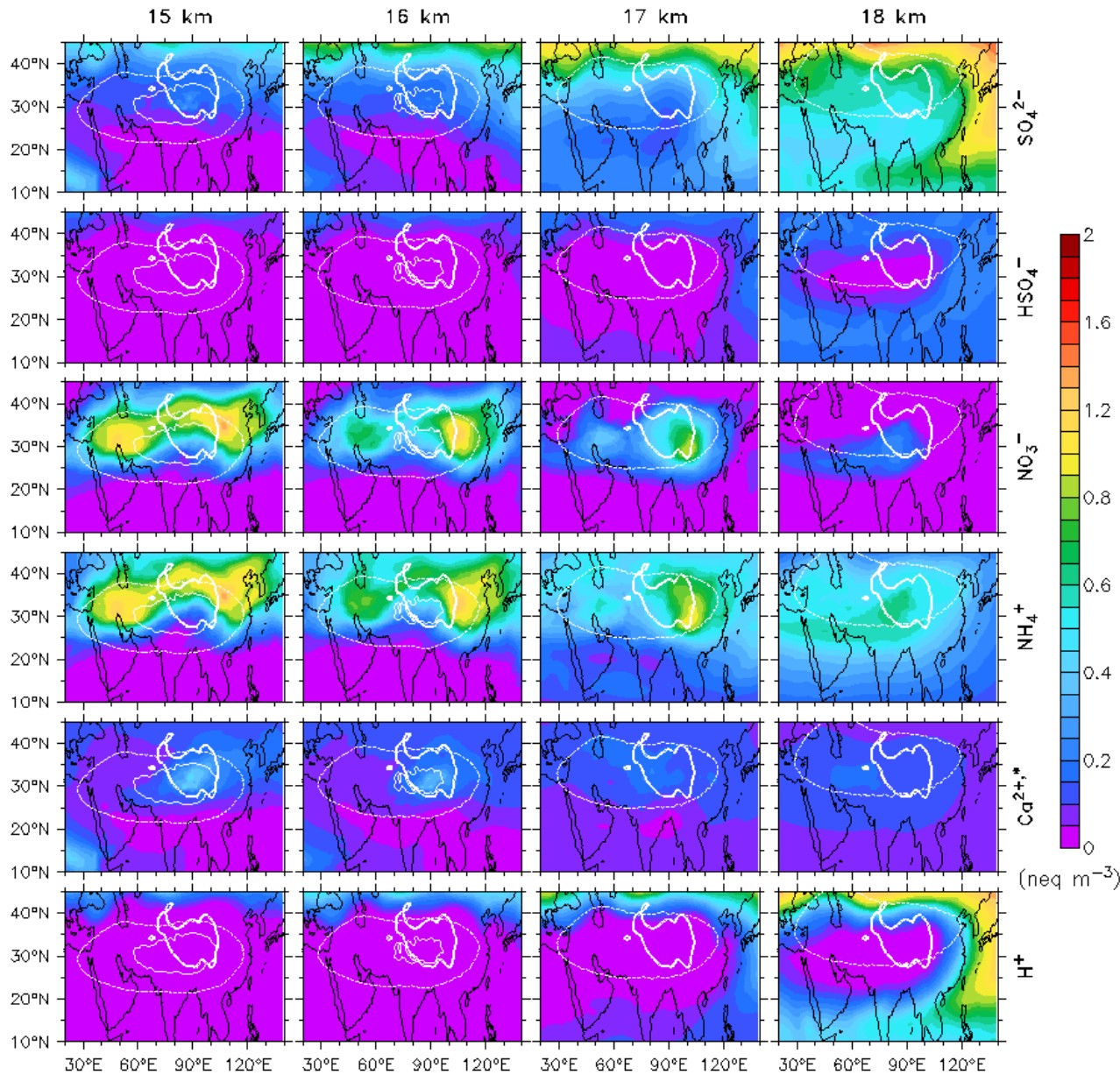

**Figure 9.** Same as Fig. 8, but for the year 2012 with a different colour scale.

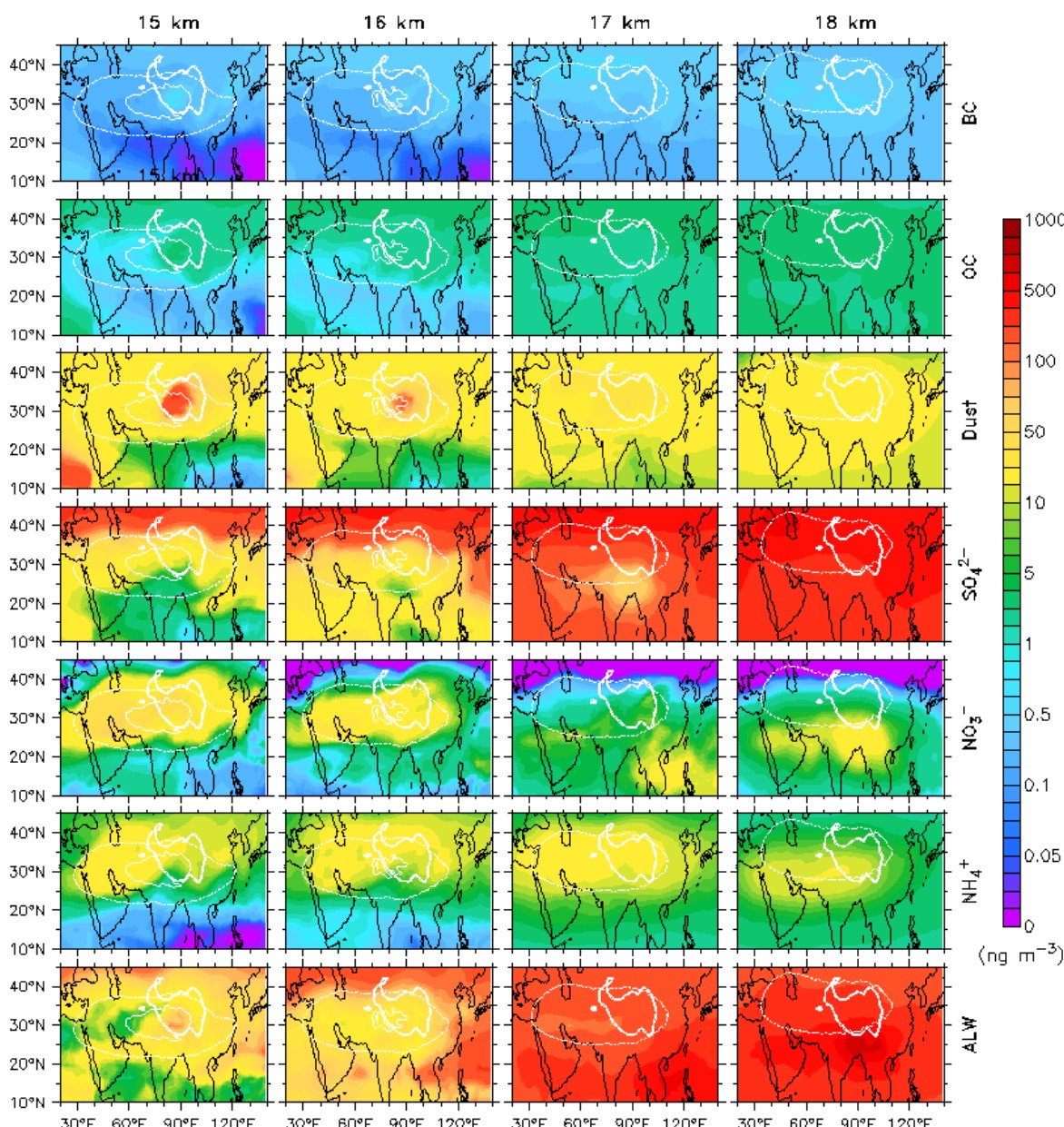

**Figure 10.** EMAC simulated mass concentrations of aerosols in the accumulation mode, including BC, OC, mineral dust, sulfate ($SO_4^{2-}$ plus $HSO_4^-$, denoted as $SO_4^{2-}$), nitrate ($NO_3^-$), ammonium ($NH_4^+$), and ALW (**first to seventh row**) in units of nanogram per cubic meter of air (ng m$^{-3}$), at altitudes of 15, 16, 17 and 18 km a.s.l. (**first to forth column**), averaged for July–August 2011. Thin white lines indicate the anticyclone area, with the same index as used in Fig. 2, and thick white lines highlight the Tibetan Plateau area.

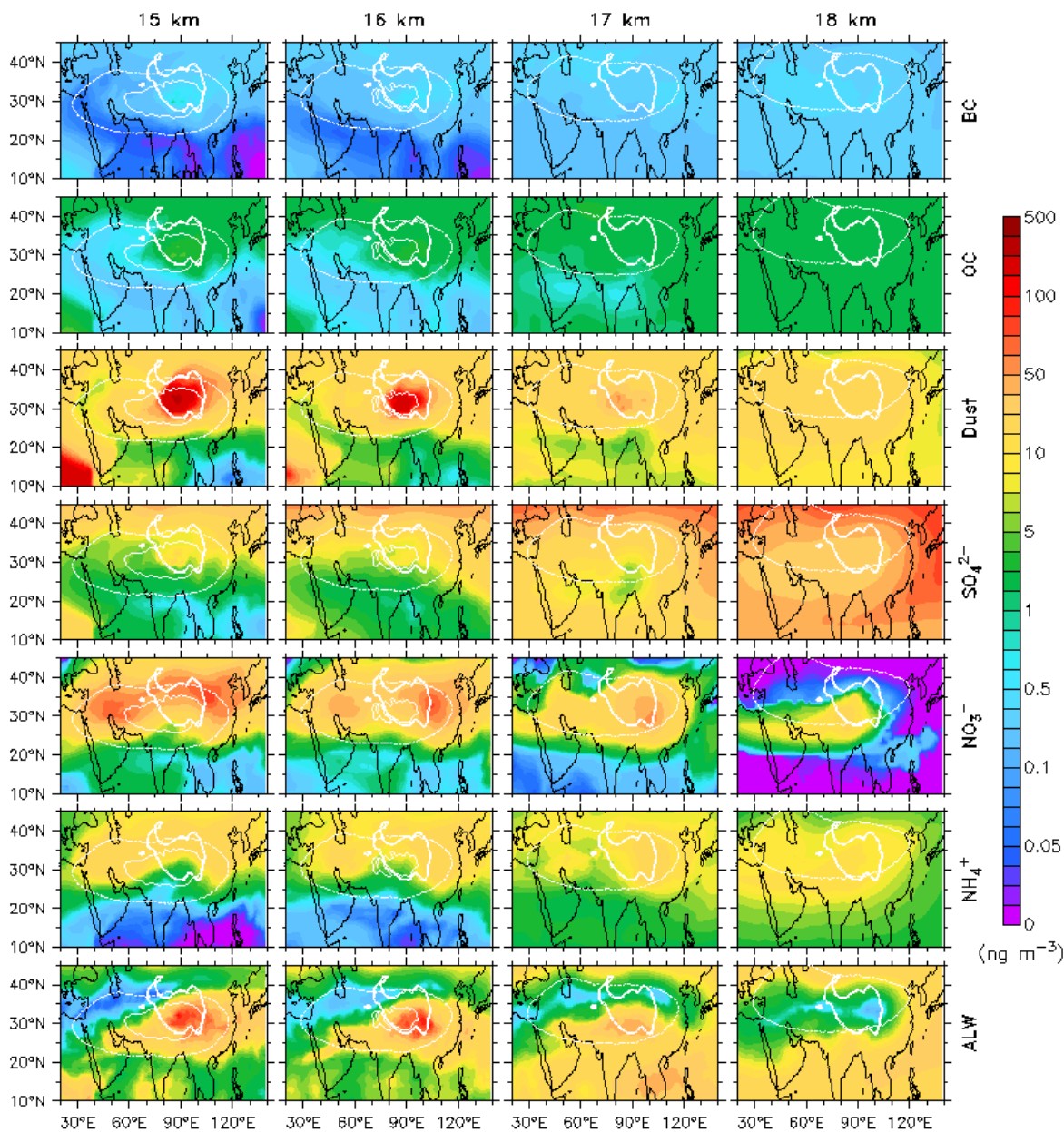

**Figure 11.** Same as Fig. 10, but for the year 2012 with a different colour scale.

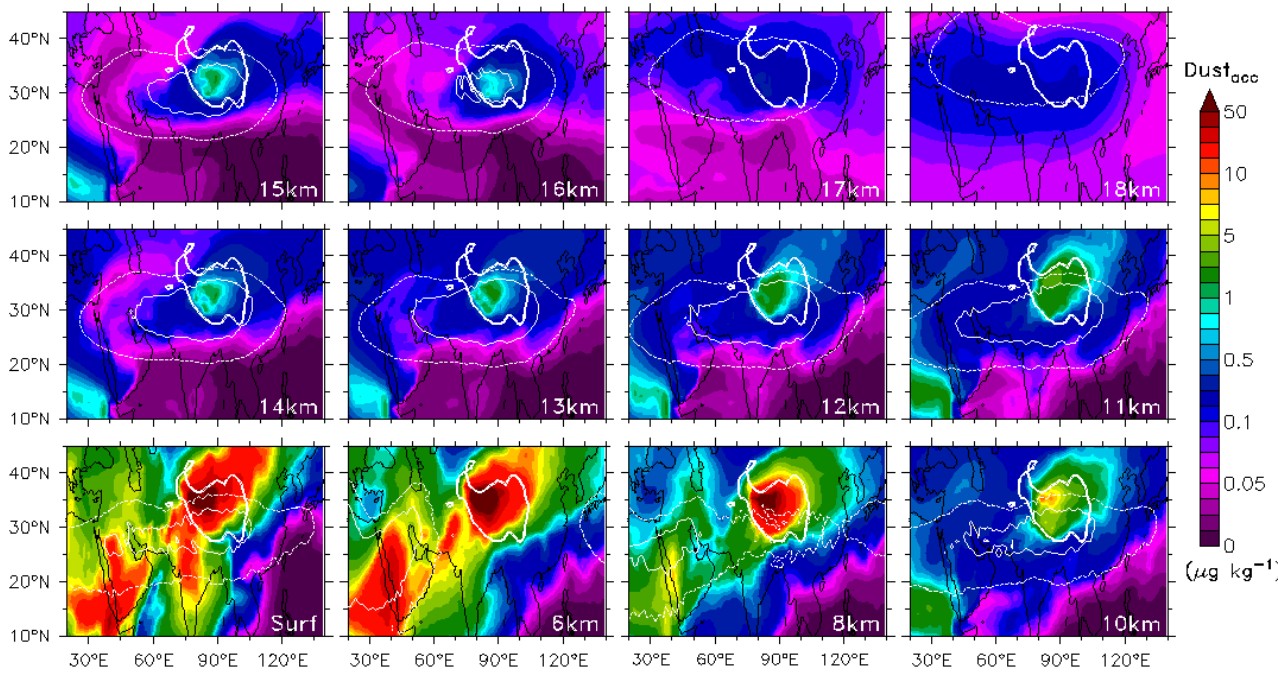

**Figure 12.** EMAC simulated dust mass concentrations in the accumulation mode, Dust$_{acc}$, in units of microgram of dust per kilogram of air (µg kg$^{-1}$) (note that a molecular weight of 40.08 is used for dust), at selected altitudes, i.e. at the surface, 6 km, 8km, 10 km, 11 km, 12 km, 13 km, 14 km, 15km, 16 km, 17 km and 18 km a.s.l., averaged for July–August 2012. Thin white lines indicate refers to the contour of pressure deviation, as defined in Fig. 2, and thick white lines highlight the Tibetan Plateau area.

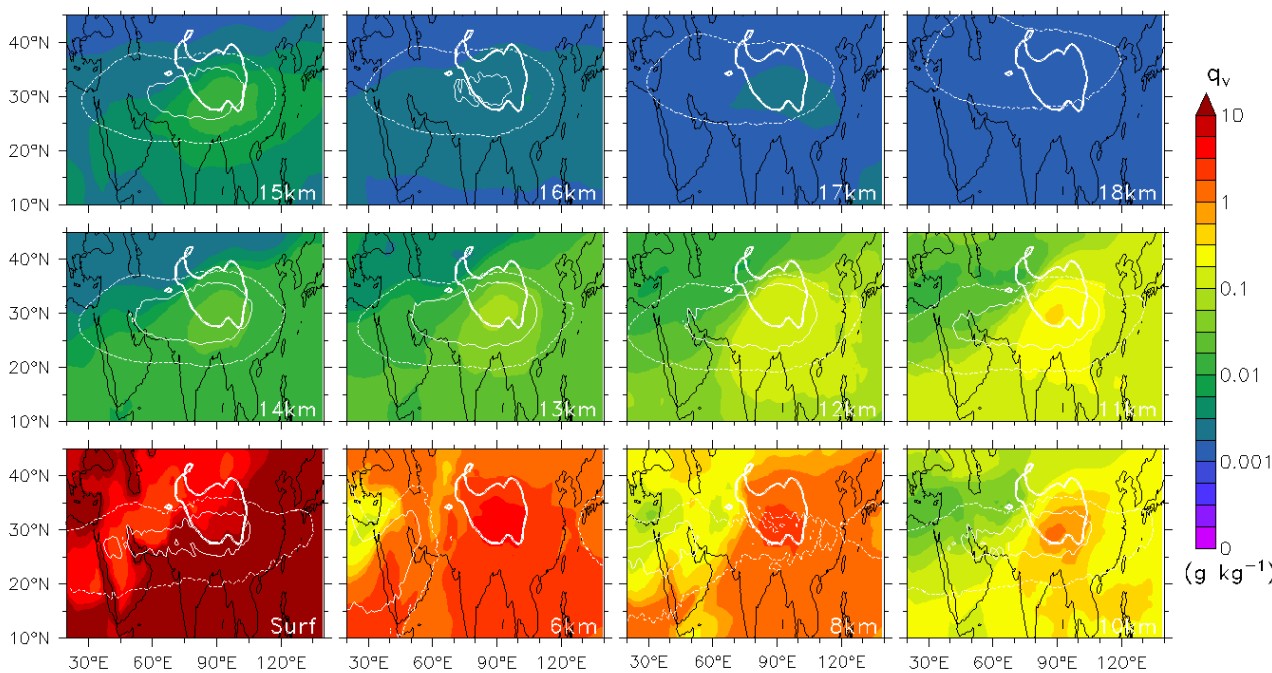

**Figure 13.** EMAC simulated specific humidity, $q_v$, in units of gram of water vapor per kilogram of air (g kg$^{-1}$), at selected altitudes, i.e. at the surface, 6 km, 8km, 10 km, 11 km, 12 km, 13 km, 14 km, 15km, 16 km, 17 km and 18 km a.s.l., averaged for July–August 2012. Thin white lines indicate refers to the contour of pressure deviation, as defined in Fig. 2, and thick white lines highlight the Tibetan Plateau area.

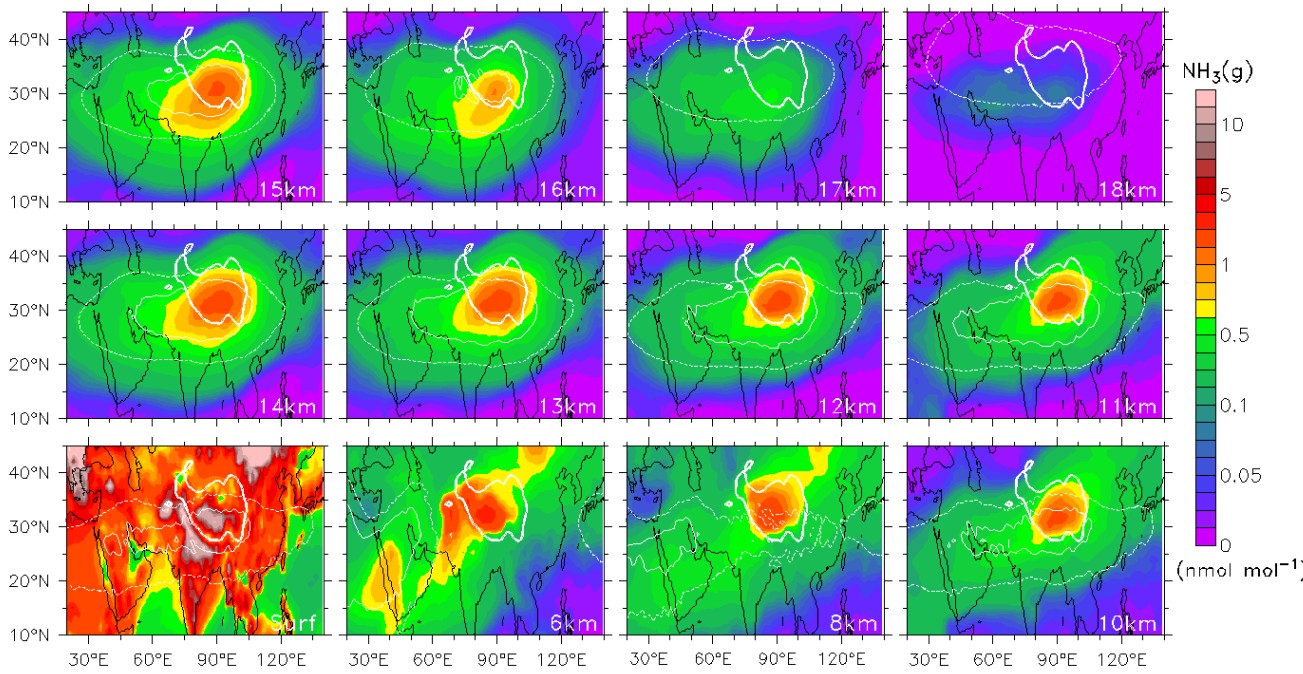

**Figure 14.** EMAC simulated ammonia in the gasphase, $NH_3$ (g), in units of nanomole of ammonia per mol of air (nmol mol[-1]) or parts of amonin per billion of air in volume (ppbv), at selected altitudes, i.e. at the surface, 6 km, 8km, 10 km, 11 km, 12 km, 13 km, 14 km, 15km, 16 km, 17 km and 18 km a.s.l., averaged for July–August 2012. Thin white lines indicate refers to the contour of pressure deviation, as defined in Fig. 2, and thick white lines highlight the Tibetan Plateau area.