# Peer review of "Modelling the aerosol chemical composition of the tropopause over the Tibetan Plateau during the Asian summer monsoon"

_Atmospheric Chemistry and Physics, 2019_

## Referee Comment (RC1) · Anonymous Referee #1 · 3 Jun 2019

**General comments**

Ma et al. study the composition of the Asian Tropopause Aerosol Layer (ATAL), a feature associated with the upper tropospheric Asian Summer Monsoon (ASM) anticyclone. The ASM, and particularly ATAL, has received quite some attention recently. Timely publication of this paper might e.g. inform upcoming studies from the StratoClim project (http://www.stratoclim.org/), which addresses similar research questions.

The paper is based on a high resolution global simulation with a detailed representation of gas phase chemistry, aerosols and particularly mineral dust. Previous studies disagree on the ATAL composition. Ma et al. highlight the importance of the latter, also

discussing the source areas. The methodology is sound, the paper nicely complements previous studies, it is very well written and well suited for ACP(D).

Before publication in ACP the following three aspects need to be addressed, which might or might not need major revisions:

(1) From the references provided it is not clear to me, if the used or a similar setup has ever been evaluated for the EMAC model. If there is such an evaluation, please specifically refer to it. If there is no such evaluation of the T106L90 resolution with chemistry yet, a separate evaluation paper (e.g. in GMD) might be considered - or at least a dedicated supplement. The solitary comparison to a measured profile of aerosol extinction coefficients is just a start. Furthermore, there are a lot of comparative statements about the simulation in the text. It is often not clear, whether those refer to observations, other simulations, or something else. This needs to be clarified, e.g. by an explicit evaluation against observations or other simulations.

(2) Convection and the associated emission of lightning NOx plays a central role in the ASM anticyclone, affecting also nitrate aerosols. Convection and lightning NOx emissions are known to depend on model resolution. How does the simulation perform with respect to observed convection? Are the lightning NOx emissions in a reasonable range, globally and within the anticyclone? Those are notorious uncertainties in modelling, but the discussion of the relative importance of nitrate aerosols depends on the availability of NOx.

(3) Please consider to add a statement about data availability (model code, setup).

**Specific comments**

In the following URLs or full citations are only given for references not provided in the draft. Otherwise please refer to the references' list in Ma et al.

P1L23: emissions, chemistry and *transport*

P2L3: Isn't the UTLS rather defined by the exchange processes between stratosphere and troposphere than by some exact distance to the (which?) tropopause? Please discuss or specifically provide a citation for that definition.

P2L15: The ASM anticyclone is not only driven by deep convection, but also by the interaction of orography with a sea breeze and heating of the Tibetan plateau. No need to discuss in detail. Just appreciate the complexity, particularly since your simulation represents topography better than coarser simulations (see P4L18).

P4L18: The Jöckel et al. papers often refer to T42L90, which seems to be some kind of standard resolution for EMAC and should also be mentioned here.

P5L1: If there is an evaluation for T106L90, please specifically refer to that.

P5L5: Consider adding Jöckel et al. (2016) to the listing of EMAC model evaluation studies.

P5L9: Consider to move the statement about spin-up from the end of the section to here.

P5L11: Please provide more details about the chemical mechanism or provide a citation to make it reproducible.

P6L7: Isn't the aerosol just facilitating gas phase reactions in this context, i.e. educts and products are both gaseous?

P6L23: Is there something like a "DLR-MACCity" inventory? Is this MACCity with species redistributed to the chemical mechanism used by Jöckel et al. (2016)?

P6L25: Please elaborate to avoid quotation marks. Is the worst case scenario the best match for real emissions in that period?

P6L28: Lightning emissions depend on model resolution and parameterizations contain tuning factors. How big is your annual global lightning NOx emissions or towards which value did you tune? Please consider adding a more thorough analysis and dis-

cussion of lightning NOx, because this is a crucial aspect of atmospheric composition in the ASM anticyclone (Barret et al., 2016; Gottschaldt et al., 2018) and affects nitrate aerosols.

P7L16: Is "crustal species" = minerals of the Earth's crust? Why do the given species not sum up to 100 percent?

P7L21: Do you use the same inventory for volcanic SO2 as Brühl et al (2018)?

P7L34: Were there any major eruptions that should have been included?

P8L9: Is this one of the nudging profiles used by Jöckel et al. (2016)?

P8L24: "regionally averaged" ... Which region? The one covered by fig. 2?

P8L27: There is eddy shedding to the east and to the west of the ASM anticyclone.

P8L31: "well represented" ... Such a statement needs comparisons to observations.

P9L56: Those studies did not use EMAC, did they? Consider rephrasing.

P9L8: Why not showing the average of multiple years? Is 2011 somehow representative for the climatologigal mean state?

P9L18: Consider scanning the literature for results of the StratoClim campaign(s). For instance, Brunamonti et al. 2018 (https://www.atmos-chem-phys.net/18/15937/2018/) show aerosol measurements within the ASM anticyclone.

P9L33: Consider putting this into perspective by discussing differences between 2011 and 2015.

P10L5: "reproduced" ... To which observations do you refer here?

P10L9: Are there observations for 2011, showing a similar anti-ATAL effect?

P10L32: Why do you focus on 2011, given that it is an unusual year for ATAL?

P11L22: This is only true without dynamic instabilities (see e.g. Gottschaldt et al.,

2018). Please consider a more differentiated formulation.

P11L31: This statement seems to contradict the previous statement about blocking (P11L22).

P12L13: Is there a reason ror not using isentropic coordinates, which should better reflect lateral transport?

P12L15: Consider rephrasing the rationale for not showing those: Ok for comparisons to corresponding observations, but distinction might make sense for analysing model results only.

P12L26: Do you refer to the interior or the surroundings of the ASM anticyclone here?

P13L7: It's probably not only due to the dust scheme, but also to the high resolution.

P13L10: Did Fadnavis et al. not consider volcanic sulfate, e.g. via some inventory?

P13L18: Such a striking discrepancy deserves elaboration. A thorough discussion of the differences might be difficult, but are there any ideas to explain this?

P13L23: This is much less than simulated by EMAC (fig. 10, 11) -> Reasons for this discrepancy?

P14L4: Consider a more cautious formulation here, because there is no consensus with other studies.

P14L19: Tagging the emission regions would help here. However, the connection of the plume over the Tibetan plateau to the surface is convincing. Consider to substitute "spatial distribution" by a more explicit formulation.

P14L23: This is strange, because near surface concentrations in Fig. 12 are not increased north of the conduit. However, it's hard to judge from fig. 3. The apparent shift to the north could just be an artefact of the colour scale and the white line over the maximum in Fig. 3. Please clarify.

P15L15-17: Didn't Brühl et al. just consider T106L31? Please try to disentangle the effects of vertical and horizontal resolution more clearly.

P15L25-28: Do you recommend T106L90 for this type of study in general OR only if there was some improvements to the convection scheme?

P15L25-28: T63L90 fits best to observations, because the convection scheme was developed for this resolution. Is this correct?

P15L25-28: You claim that T106L90 is better for transport over complex terrain. That is intuitively clear, but strictly would need to be shown. However, what is dominating transport to the UTLS: convection or orographic forcing from the terrain?

P15L25-28: Another point to consider is whether or not the occurrence of convective events is represented more realistically by T63 or T106. Please discuss those aspects to better justify your recommendation for T106L90. Consider comparing convection or lightning activity to observations to get a better idea of how realistically your simulation is (see also General comments).

P15L32: "improve" . . . Compared to what?

P16L1: "enhance" . . . Compared to what?

P28L4: Consider using SI units

P30L3: Confusing description of the quantity: Do you mean just the tropospheric burden?

**Technical corrections**

P31, Fig. 4: Figure blurry -> consider higher resolution or a vector graphics format
* * *

---

## Author Comment (AC1) · 17 Jul 2019

The comment was uploaded in the form of a supplement:
https://www.atmos-chem-phys-discuss.net/acp-2019-412/acp-2019-412-AC1-supplement.pdf

---

## Referee Comment (RC2) · Anonymous Referee #2 · 31 Jul 2019

General Comments:

This interesting work studies the aerosol chemical composition in the Asian Tropopause Aerosol Layer (ATAL), and uses EMAC model running at a high resolution to investigate the links between ATAL and Asian Summer Monsoon (ASM). The aerosol properties in the ATAL have been the subject of discussion over the past decade and have received quite some attention recently. I agree with the Reviewer#1 that the results of this work nicely complement previous studies. The manuscript is well written, the methodology is detailed described and sound, and the results are well presented and discussed. The authors have addressed most of my concerns in the response to

Reviewer#1. I feel this work is suitable for publication in ACP after address the following minor concerns.

Specific comments:

1) It would be best to rephase some expressions in the manuscript, considering the similarity index of 23% (see Similarity Report). No doubt regarding the originality of this work, which will be mentioned in my later comments, but I feel some re-wording to credit the previous works in a better way would help improve this manuscript.

2) About the altitudes of 15-18 km. In order to avoid confusion, please find a suitable place to clearly state that whether it is above the sea level or over ground.

3) P5L24. I am wondering that how is organic aerosol formation simulated in the ORA-CLE sub-model, if partitioning of secondary organic aerosol between gas and particle phases is not considered? Some elaboration may be needed here.

4) P10L19. The acronym 'SS' for sea spray. Please place it at the first time when 'sea spray' was used.

5) P10L32. The contribution of WASO and ALW to aerosol extinction is much higher in 2011 than in 2010/2012. Is it due to the pronounced Nabro eruption in 2011? This eruption may enhance the highly hygroscopic components in UTLS such as sulfate originated from SO2, and enlarge the contributions from WASO and ALW? Some discussion about the difference between 2011 and 2010/2012 would be interesting to see.

6) P11L11. The definition of 'Ca2+*' needs to be clarified. Cations of Na, K, Mg and Ca are accounted for as 'Ca2+*'. Here, you mean by mass, mol, or charge balancing? Furthermore, Na+ is also a typical tracer for sea salt. Drop sodium and just sum up calcium, magnesium and calcium. Would this be a better tracer for mineral dust?

7) P13L27. Typo? 'ACM' change to ASM ?

8) Section 3.5. A nice discussion to figure out the source of dust in the UTLS by

holographic scanning the transport pathway from surface to 18 km. My first suggestion is: would it be more generally representative to show the results of 2010/2012, but place the results of 2011 in supplement? Anyway, it would not change the results significantly, because 'transport patterns are the same in 2010 and 2012 as in 2011'; however, 2011 is a special year tagged by the pronounced Nabro eruption with potential impacts on climate the general circulation, which may lead to unnecessary doubt from audience about its representative. The second suggestion is: would it be possible to have some discussion about the source or transport pathway of WASO and ALW (or water vapor), which also considerably contribute to aerosol extinction coefficient in the ATAL?

9) P17L9. I am curious that why finer resolution (T106L90) underestimate the convective vertical transport, but not the coarser resolution. Should the finer resolution represent convective processes better, although none of them explicitly describe the convective processes? Would you help me understand it.

10) The conclusion. I feel add some clear statement to highlight the originality and novelty of this work would help audience get the full picture of this nice study. In the introduction and results discussion, authors made extensive comparisons with previous studies to evaluate the results of this study. Add some sentences to clearly state the new findings or improvements of this study will be helpful. For example, including nitrate which is missing in lots of previous studies, find more dust contribution by updating dust emission scheme and etc.

---

## Author Comment (AC2) · 15 Aug 2019

The comment was uploaded in the form of a supplement:
https://www.atmos-chem-phys-discuss.net/acp-2019-412/acp-2019-412-AC2-supplement.pdf

---

## Author Response (AR1)

We would like to thank the two referees for their insightful and constructive comments.

**Reply to Anonymous Referee #1**

Referee comments are in black. Author responses are in blue.

**General comments**

Ma et al. study the composition of the Asian Tropopause Aerosol Layer (ATAL), a feature associated with the upper tropospheric Asian Summer Monsoon (ASM) anticyclone. The ASM, and particularly ATAL, has received quite some attention recently. Timely publication of this paper might e.g. inform upcoming studies from the StratoClim project (http://www.stratoclim.org/), which addresses similar research questions.

The paper is based on a high resolution global simulation with a detailed representation of gas phase chemistry, aerosols and particularly mineral dust. Previous studies disagree on the ATAL composition. Ma et al. highlight the importance of the latter, also discussing the source areas. The methodology is sound, the paper nicely complements previous studies, it is very well written and well suited for ACP(D).

Before publication in ACP the following three aspects need to be addressed, which might or might not need major revisions:

We thank the anonymous referee for his/her insightful comments. Below are our point-by-point responses to referee's comments in detail.

(1) From the references provided it is not clear to me, if the used or a similar setup has ever been evaluated for the EMAC model. If there is such an evaluation, please specifically refer to it. If there is no such evaluation of the T106L90 resolution with chemistry yet, a separate evaluation paper (e.g. in GMD) might be considered – or at least a dedicated supplement. The solitary comparison to a measured profile of aerosol extinction coefficients is just a start. Furthermore, there are a lot of comparative statements about the simulation in the text. It is often not clear, whether those refer to observations, other simulations, or something else. This needs to be clarified, e.g. by an

explicit evaluation against observations or other simulations.

EMAC is a very complicated model system coupled with different modules to simulate various physical and chemical processes simultaneously in both the troposphere and stratosphere. The model is updated and improved, by introducing new and advanced modules continuously. The references we provided in the first paragraph of Sect. 2 describe these updates and improvements as well as evaluations against observations. In subsequent paragraphs, we describe the model setup, i.e., modules adopted in this study, such as MECCA, JVAL, GMXe, ISORROPIA-II, MECCA_KHET, AEROPT, SCAV, DRYDEP and so on, with their references provided. The standard versions of EMAC introduced by Jöckel et al. (2006, 2010, 2016) and its modules, e.g. GMXe (Pringle et al., 2010), were evaluated for a typical horizontal resolution of T42. Indeed, there were several studies using the T106 resolution and, as suggested by the reviewer, we have added some sentences to the end of Sect. 2. to describe them in detail:

"*Pozzer et al. (2012) used the same modules (e.g., GMXe and ISORROPIA-II for aerosols) as in this study but different emission data (e.g., dust emissions from AEROCOM) in the T106L31 simulations of tropospheric aerosols for the years 2005-2008. By comparing the model output with observations from different measurement networks and satellite remote sensing instruments, they found that the main spatial and temporal atmospheric distribution of sulfate, ammonium and nitrate aerosols were well reproduced in general, but there was an underestimation of aeolian dust emissions in the dust outflow regions (Pozzer et al., 2012). Klingmüller et al. (2018) develop an advanced dust emission scheme, with the updated land cover classification, the inclusion of the topography factor and the modification of the sandblasting efficiency function. They performed the T106L31 simulations for the year 2011; by comparing with the aerosol optical depth (AOD), dust concentrations and deposition results from various observational platforms, they concluded that the update significantly improves agreement with the observations (Klingmüller et al., 2018). With respect to aerosol and stratospheric chemistry, the model setup in this study is similar to those used in Brühl et al. (2015, 2018), the latter of which (Brühl et al.,2018) used the T106L90 resolution for a 1-year sensitivity test*".

Theoretically, chemical processes simulated with state-of-science chemical reactions in the models like EMAC should be independent of model resolution. However, due to large gradients in emission rates of its precursors, e.g., around megacities, and non-linear ozone chemistry, simulated ozone in some regions might be affected by model resolution (e.g., Kentarchos et al., 2001;Stock et al., 2014). Wild and Prather (2006) showed that compared to the observations, the simulated ozone at T21, T42, T63 and T106 is increasingly realistic with resolution. Even so, non-linear ozone chemistry associated with pollution emissions and its effect on aerosols should not be so significant in remote regions, in particular at the tropopause. We think that dynamical processes that are strongly affected by the topography of the Tibetan Plateau and simulated by state-of-art parameterizations in the model, such as convection and

lightning as pointed out by the reviewer in Question 2, and associated chemical and physical processes may have a larger resolution effect on the simulation of the ATAL. A full evaluation of the T106L90 with chemistry, as suggested by the reviewer, can be considered in other paper(s).

5   With regards to unclear comparative statements in the text, we have revised those statements accordingly, as described in our response to specific questions below.

(2) Convection and the associated emission of lightning NOx plays a central role in the ASM anticyclone, affecting also nitrate aerosols. Convection and lightning NOx emissions are known to depend on model resolution. How does the simulation perform with respect to observed convection? Are the lightning NOx emissions in a reasonable range, globally and within the anticyclone? Those are notorious uncertainties in modelling, but the discussion of the relative importance of nitrate aerosols depends on the availability of NOx.

We agree that convection and the associated lightning $NO_x$ emissions play a central role in the $NO_x$ and OH budget in the ASM anticyclone (Gottschaldt et al., 2018; Lelieveld et al., 2018), which may have large impacts on nitrate aerosols (Tost, 2017). Both parameterizations are a notorious source of uncertainty in global models (Gottschaldt et al., 2018). As suggested by the reviewer, we have added the following paragraph in the middle part of Sect. 2 to describe the parameterization schemes for convection and lightning $NO_x$ emissions used in our simulations:

*"Convection is parameterized by the submodel CONVECT, in which the Tiedtke (1989) scheme with modifications by Nordeng (1994) is set as default (Tost et al., 2006b) and used in this study. This scheme was developed for resolution T63. As discussed in Brühl et al. (2018), it overestimates vertical transport in the UTLS for T42 and underestimates it for T106. The $NO_x$ emissions from lightning activity are calculated online using the submodel LNOX (Tost et al., 2007b). In this study we apply the parameterization developed by Grewe et al. (2001), which links the flash frequency to the updraft velocity. Here flash frequency obtained by this parameterization is scaled by a factor of 0.0695 for T106L90 simulations. In this study global emissions of $NO_x$ from lightning are simulated to be 7.9, 6.7 and 6.3 Tg (N) $a^{-1}$ for the years 2010, 2011 and 2012, respectively, falling into the range of 2–8 Tg (N) $a^{-1}$ suggested by Schumann and Huntrieser (2007). The parameterisation schemes for convection (Tiedtke, 1989; Nordeng, 1994; Tost et al., 2006b) and lightning (Grewe et al., 2001) used in this study have been evaluated with simulations at coarser resolution (Tost et al., 2007b; Lopez, 2016; Gottschaldt et al., 2018) and the agreement is particularly noticeable for the ASM anticyclone region (Gottschaldt et al., 2018)".*

We have also added the following sentences at the end of Sect. 3.3 to highlight the important role of lightning over the Tibetan Plateau and its potential effects on nitrate

aerosols in the UTLS.

*"As demonstrated in previous studies, lightning $NO_x$ clearly dominates the $NO_x$ budget from the tropopause to 100 hPa below it, and its emissions are much stronger in the Tibetan part (Gottschaldt et al., 2018). Lightning $NO_x$ also plays a central role in sustaining upper tropospheric OH concentrations over the monsoon (Lelieveld et al., 2018). In our simulations, lightning $NO_x$ emissions within the ASM anticyclone (20-140 °E and 10-45 °N) are very intensive, with estimated values of 1,1, 0.5 and 0.9 Tg (N) $a^{-1}$ in the years 2010, 2011 and 2012, respectively. High $NO_x$ and OH concentrations are in favour of $HNO_3$ formation via the reaction $NO_2$ + OH, which may provide an import source of nitrate within the anticyclone. Tost (2017) found 60% of aerosol nitrate between 500 hPa and the tropopause being produced from lightning in the ASM and its outflow under the present climatic condition. In addition to the available nitric acid, other factors such as neutralising ions (e.g., $Ca^{2+}$, $NH_4^+$ and $SO_4^{2-}$) and temperature can also influence the amount of nitrate aerosols in the ATAL. How these factors and associated processes affect nitrate aerosols in the UTLS over the Tibetan Plateau will be investigated thoroughly in a future study".*

(3) Please consider to add a statement about data availability (model code, setup).
We have added the statements about Code and data availability, Competing interests and Special issue in the revised manuscript.

**Specific comments**

In the following URLs or full citations are only given for references not provided in the draft. Otherwise please refer to the references' list in Ma et al.
So do for our responses.

P1L23: emissions, chemistry and transport
Done.

P2L3: Isn't the UTLS rather defined by the exchange processes between stratosphere and troposphere than by some exact distance to the (which?) tropopause? Please discuss or specifically provide a citation for that definition.
We agree that it seems better to define the UTLS by exchange processes rather than by some exact distance. Here we use a rough distance value from the work (2nd sentence in Introduction) of Gettelman et al. (2011) merely to give a concept about the possible range of the ULS. The phrase has been changed to "*ranging about ±5 km around the tropopause (Gettelman et al., 2011)*" in the revised manuscript.

P2L15: The ASM anticyclone is not only driven by deep convection, but also by the interaction of orography with a sea breeze and heating of the Tibetan plateau. No need to discuss in detail. Just appreciate the complexity, particularly since your simulation represents topography better than coarser simulations (see P4L18).

We agree. In the beginning of Sect. 3.1, we state "*The ASM circulation is characterized by cyclonic flow and convergence in the lower troposphere, and a strong anticyclone and divergence in the UTLS (Krishnamurti and Bhalme, 1976); its structure is primarily a response to diabatic heating associated with deep convection (Hoskins and Rodwell, 1995). Known as the "sensible heat pump", the Tibetan Plateau modifies monsoon circulation (Wu and Zhang, 1998) and even modulates large-scale atmospheric circulations over the Northern Hemisphere (Zhao and Chen, 2001;Zhou et al., 2009)*" .

P4L18: The Jöckel et al. papers often refer to T42L90, which seems to be some kind of standard resolution for EMAC and should also be mentioned here.
Done.

P5L1: If there is an evaluation for T106L90, please specifically refer to that.
As stated in our response to General comments (Q1) above, we have specified the evaluations for T106L90 in the end of Sect. 2. of the revised manuscript: "*Pozzer et al. (2012) used the same modules ……the T106L90 resolution for a 1-year sensitivity test*".

P5L5: Consider adding Jöckel et al. (2016) to the listing of EMAC model evaluation studies.
Done.

P5L9: Consider to move the statement about spin-up from the end of the section to here.
Done.

P5L11: Please provide more details about the chemical mechanism or provide a citation to make it reproducible.
We have added the following sentence in the revised manuscript: "*The chemical mechanism used in this study is primarily based on stratospheric chemistry used by Brühl et al. (2015) plus VOC chemistry reported by Taraborrelli et al. (2012)*".

P6L7: Isn't the aerosol just facilitating gas phase reactions in this context, i.e. educts and products are both gaseous?
In some cases, the aerosol is just facilitating gas-phase reactions. In other cases, educts and products are not released as gases, and instead they are kept in the aerosol, either in solid or liquid state. These depend on what reaction it is and what region (the stratosphere or the troposphere) the reaction takes place. For example, for the reaction $N_2O_5 + H_2O \rightarrow HNO_3$ on aerosol surfaces, $N_2O_5$ is converted into aqueous-phase nitrate in the troposphere, whereas in the dry stratosphere, gaseous $HNO_3$ is produced. For detailed information, please see Sect. 7.1.2 MECCA KHET in the work of Jöckel et al. (2010).

P6L23: Is there something like a "DLR-MACCity" inventory? Is this MACCity with species redistributed to the chemical mechanism used by Jöckel et al. (2016)?
It is actually the same inventory as reported by Jöckel et al. (2016). To avoid confusion, we have removed "*DLR-*" in the revised manuscript.

P6L25: Please elaborate to avoid quotation marks. Is the worst case scenario the best match for real emissions in that period?

The quotation mask has been removed in the revised manuscript. It is an estimate of historical emissions, instead of projected emissions based on different scenarios. Maybe we misunderstand something. For the period 2005 to 2015 the worst case scenario matches observations best for most species.

P6L28: Lightning emissions depend on model resolution and parameterizations contain tuning factors. How big is your annual global lightning NOx emissions or towards which value did you tune? Please consider adding a more thorough analysis and discussion of lightning NOx, because this is a crucial aspect of atmospheric composition in the ASM anticyclone (Barret et al., 2016; Gottschaldt et al., 2018) and affects nitrate aerosols.

As stated in our response to General comments (Q2) above, the global $NO_x$ emissions from lightning are simulated to be 7.9, 6.7 and 6.3 Tg (N) a$^{-1}$ for the years 2010, 2011 and 2012, respectively, falling into the range of 2–8 Tg (N) a$^{-1}$ suggested by Schumann and Huntrieser (2007). We have added some sentences at the end of Sect. 3.3 to highlight the important role of lightning over the Tibetan Plateau and its potential effects on nitrate aerosols in the UTLS. It should be noted that the formation of nitrate aerosols and partitioning of nitrate between gaseous and aerosol phases are complicated, depending on other more factors (e.g., gaseous $NH_3$, pH of aerosol liquid solution, and temperature) than gaseous $NO_x$ (specifically $HNO_3$), and will be investigated thoroughly in another paper in the near future.

P7L16: Is "crustal species" = minerals of the Earth's crust? Why do the given species not sum up to 100 percent?

Yes, here we mean species in selected ionic form. They sum up not to 100% because the rest exist in other form, which are dominant and still treated as chemical inert dust in the model.

P7L21: Do you use the same inventory for volcanic SO2 as Brühl et al (2018)?

Yes, we do. To avoid confusion, we have changed the phrase to "*As in the work of Brühl et al. (2018),*" in the revised manuscript.

P7L34: Were there any major eruptions that should have been included?

We have added the following state in the revised manuscript: "*There were two large major volcanic eruptions occurring in 2012, namely Soputan (124.73 °E, 1.11 °N) on 18 September 2012 and Copahue (288.8 °E, 37.86 °S) on 22 December 2012, with the amounts of ejected SO$_2$ being 1-2 orders of magnitude smaller than that of Nabro, as reported by Mills et al. (2016). These two volcanic eruptions are not included for the simulations of this study*". Since they occurred after the ASM period we are investigating, neglecting them in the simulations should have not any effects on the results for this study.

P8L9: Is this one of the nudging profiles used by Jöckel et al. (2016)?

Yes. The nudging profile should be the same as demonstrated in Fig. S5 of Jöckel et al. (2016). We have added the phrase "*as used by Jöckel et al. (2016)*" in the revised manuscript.

P8L24: "regionally averaged" … Which region? The one covered by fig. 2?
It is the region covered by Fig. 2. We have added the phrase "*over 20–140 ºE and 10–45 ºN*" in the revised manuscript.

P8L27: There is eddy shedding to the east and to the west of the ASM anticyclone.
We have added this sentence to the revised manuscript as suggested by the reviewer.

P8L31: "well represented" … Such a statement needs comparisons to observations.
Here we mean the general or climatic horizontal distributions of the wind field, $O_3$ and CO associated with the ASM anticyclone. In the end of this paragraph, we cited other model studies (i.e., Gettelman et al., 2004; Park et al., 2007; Park et al., 2009; Barret et al., 2016; Santee et al., 2017) for such $O_3$ and CO distributions. We also have added the following sentence in the revised manuscript: "*It was shown recently that EMAC realistically simulates reactive gases and radicals within the ASM anticyclone by comparing the simulation results with aircraft measurements (Lelieveld et al., 2018)*".

P9L5-6: Those studies did not use EMAC, did they? Consider rephrasing.
No, they didn't. The phrase has been changed to "*This feature has been well reproduced by EMAC using $O_3$ as a stratospheric tracer and CO as a tropospheric tracer as also done in other model studies*".

P9L8: Why not showing the average of multiple years? Is 2011 somehow representative for the climatological mean state?
By comparing Fig. 3 with Fig. S1 and Fig. S2, one can see nearly the same global distributions of these aerosol columns (e.g., dust) among 2010, 2011 and 2012. So the year 2011 is representative for the climatological mean state in the early 2010s. We show the three years of aerosol columns separately considering that one may want to look at a specific year for comparison with aerosols in the UTLS, which are investigated year on year.

P9L18: Consider scanning the literature for results of the StratoClim campaign(s). For instance, Brunamonti et al. 2018 (https://www.atmos-chem-phys.net/18/15937/2018/) show aerosol measurements within the ASM anticyclone.
We have added the following statement in the revised manuscript "*By balloon-borne measurements, Brunamonti et al. (2018) found the maximum aerosol backscatter occurring at the cold-point tropopause, revealing the thermodynamically significant levels of the ATAL*".

P9L33: Consider putting this into perspective by discussing differences between 2011and 2015.
Do you mean P10L33 and between 2011 and 2012? We have added this into the second paragraph of the Conclusions in the revised manuscript: "*In contrast to the absolute $K_e$*

*from mineral dust, which is at the same level in the three years, the absolute $K_e$ contributed by water-soluble species (WASO) and aerosol liquid water (ALW) is much higher in 2011 than in 2010 and 2012, especially in the lower stratosphere (e.g., at 17 km)*".

P10L5: "reproduced" ... To which observations do you refer here?
We have added the phrase "*as shown by Vernier et al. (2011) and Thomason and Vernier (2013)*" in the revised manuscript.

P10L9: Are there observations for 2011, showing a similar anti-ATAL effect?
We did not find any literature reporting such observations for 2011.

P10L32: Why do you focus on 2011, given that it is an unusual year for ATAL?
Here we try to explain this phenomenon caused by the volcanic eruption, which is unusual for ATAL but does happen once every few years.

P11L22: This is only true without dynamic instabilities (see e.g. Gottschaldt et al., 2018). Please consider a more differentiated formulation.
We have changed the statement to "*The ASM anticyclone not only traps tropospheric pollutants inside but also to some extent blocks intrusions of stratospheric ozone and aerosols from outside. It should be noted that here we only investigate a seasonal mean aspect of the ASM anticyclone. Due to the dynamical instabilities of the ASM anticyclone, entrainment of stratospheric tracers does occur frequently, approximately twice a month (Gottschaldt et al., 2018)*".

P11L31: This statement seems to contradict the previous statement about blocking (P11L22).
Please see our reply above (P11L22).

P12L13: Is there a reason for not using isentropic coordinates, which should better reflect lateral transport?
Yes, we agree that isentropic coordinates should be better to reflect lateral transport. Here we use altitudes considering that they are popular for satellite products and sounding measurements. The model uses hybrid pressure coordinates.

P12L15: Consider rephrasing the rationale for not showing those: Ok for comparisons to corresponding observations, but distinction might make sense for analysing model results only.
The statement has been rephrased as "*For simplicity, the crustal species $Ca^{2+}$, $Na^+$, $K^+$ and $Mg^{2+}$ are not shown individually, considering that these ionic species within the ASM anticyclone originate predominantly from mineral dust, having the same constant relative fractions as their emission sources do (see Sect. 2 above)*".

P12L26: Do you refer to the interior or the surroundings of the ASM anticyclone here?
We refer to the interior of the ASM anticyclone, in which the Tibetan Plateau is situated.

P13L7: It's probably not only due to the dust scheme, but also to the high resolution.

We have added a statement about the resolution effect in the revised manuscript.

P13L10: Did Fadnavis et al. not consider volcanic sulfate, e.g. via some inventory?
This was not clearly described in the work of Fadnavis et al. (2013). We have rephrased the statement as "*since volcanic sulfate was most likely not considered in their simulation*" in the revised manuscript.

P13L18: Such a striking discrepancy deserves elaboration. A thorough discussion of the differences might be difficult, but are there any ideas to explain this?
We have added the following discussion in the revised manuscript: "*Similar to the comparison with Fadnavis et al. (2013) mentioned above, the different dust emission scheme and model resolution used by Yu et al. (2015) might lead to significant discrepancy of dust concentration in the UTLS within the ASM anticyclone compared to this study. Moreover, Yu et al. (2015) did not take into account stratospheric volcanic emissions and nitrate aerosols in their simulation, which might underestimate the contribution of inorganic aerosols to the ATAL with respect to this study*".

P13L23: This is much less than simulated by EMAC (fig. 10, 11) -> Reasons for this discrepancy?
Gu et al. (2016) referred Fairlie et al. (2007) for the mineral dust treatment in GEOS-Chem. Fairlie et al. (2007) reported that global dust emissions were 178 Tg a$^{-1}$ for accumulation mode particles (radius of 0.1–1.0 μm) and 1453 Tg a$^{-1}$ in total (radius at 0.1–6.0 μm) for the year 2001. The dust scheme of EMAC used in this study is based on the work of Klingmüller et al. (2018). According to Klingmüller et al. (2018), global dust emissions were 148 Tg a$^{-1}$ for accumulation mode particles and 1310 Tg a$^{-1}$ in total for the year 2011. It seems that the estimates in global dust emissions between the two models (GEOS-Chem and EMAC) are comparable. For the reasons of much less dust (mainly for the accumulation mode) in the UTLS within the ASM anticyclone, we assume that both soil-related database and topography-associated modeling strategy for the Tibetan Plateau can influence the results. These should be investigated thoroughly in future studies.

P14L4: Consider a more cautious formulation here, because there is no consensus with other studies.
We have changed the phrase "*As shown above*" to "*According to our EMAC simulations*" in the revised manuscript.

P14L19: Tagging the emission regions would help here. However, the connection of the plume over the Tibetan plateau to the surface is convincing. Consider to substitute "spatial distribution" by a more explicit formulation.
We have changed the expression to "*The plume transport pattern that connects the maximums of mineral dust aerosols over the Tibetan Plateau to the surface indicates*" in the revised manuscript.

P14L23: This is strange, because near surface concentrations in Fig. 12 are not

increased north of the conduit. However, it's hard to judge from fig. 3. The apparent shift to the north could just be an artefact of the colour scale and the white line over the maximum in Fig. 3. Please clarify.

Fig. 3 shows the tropospheric columns (mg m$^{-2}$), which are derived by integrating the aerosol loads in all model levels from the surface to the tropopause. Fig. 12 presents aerosol mass mixing ratios ($\mu$g kg$^{-1}$) at the surface and selected altitudes above sea level (a.s.l.), with the latter starting from 6 km a.s.l. If a lower altitude, e.g. 4 or 5 km a.s.l., is selected, one can see large blank areas on the plateau, simply because they are below the ground levels (see Fig. A1 below). This terrain effect can result in the shift of the column maximum to the north shown in Fig. 3.

P15L15-17: Didn't Brühl et al. just consider T106L31? Please try to disentangle the effects of vertical and horizontal resolution more clearly.

Brühl et al. (2018) considered T106L31 and T106L90.

P15L25-28: Do you recommend T106L90 for this type of study in general OR only if there was some improvements to the convection scheme?

We recommend T106L90 with improved convection parameterization. We have changed the statements to: "*Such difference might partly be attributed to the difference in simulated mineral dust, but also it is likely due to less efficiently convective transport of anthropogenic aerosols and their gaseous precursors in the T106L90 simulations compared to T63L90 simulations. It should be noted that deep convection events occur much less frequently over the northern part of the Tibetan Plateau than the southern part of it, to the latter pollution from South Asia tends to accumulate (Lelieveld et al., 2018). Our EMAC simulation at a relatively high resolution (i.e., T106L90) reveals clearly the important role of emissions and the orographic forcing in mineral dust transport over the Tibetan Plateau. T106L90 with improved convection parameterization is suggested to investigate the transport of aerosols and their gaseous precursors associated with complex topography and finer structure of the anticyclone*".

P15L25-28: T63L90 fits best to observations, because the convection scheme was developed for this resolution. Is this correct?

As described in the manuscript, the convection scheme we use for this study was developed by Tiedtke (1989) based on T63 resolution.

P15L25-28: You claim that T106L90 is better for transport over complex terrain. That is intuitively clear, but strictly would need to be shown. However, what is dominating transport to the UTLS: convection or orographic forcing from the terrain?

This should depend on the regions. On the northern part of the plateau, orographic forcing may be more important, due to less moisture and deep convection. See our reply to the first question about P15L25-28 above.

P15L25-28: Another point to consider is whether or not the occurrence of convective events is represented more realistically by T63 or T106. Please discuss those aspects to better justify your recommendation for T106L90. Consider comparing convection or

lightning activity to observations to get a better idea of how realistically your simulation is (see also General comments).

Fig. A2 shows the geographical distributions of averaged daily deep convection events for each month in 2011, from our EMAC T106L90 simulation. It seems that the distribution patterns are realistic. We also think that these comparisons are interesting, and we will do that in the near future when we have found a optimistic method to improve the convection scheme for T106L90 resolution.

P15L32: "improve" … Compared to what?
We have skipped this sentence in the revised manuscript considering that this advanced dust flux scheme has been used and tested before.

P16L1: "enhance" … Compared to what?
We have skipped this sentence in the revised manuscript considering that MIPAS $SO_2$ data was not firstly used in this study.

P28L4: Consider using SI units
Done.

P30L3: Confusing description of the quantity: Do you mean just the tropospheric burden?
Yes. The confusing phrase "*of the column concentrations*" has been deleted in the revised manuscript.

**Technical corrections**

P31, Fig. 4: Figure blurry -> consider higher resolution or a vector graphics format
The .ps files will be provided for figure production.

[Figure]

**Figure A1.** EMAC simulated dust mass concentrations in the accumulation mode, Dust$_{acc}$, in units of microgram of dust per kilogram of air (µg kg$^{-1}$) (note that a molecular weight of 40.08 is used for dust), at selected altitudes, i.e. at the surface, 1 km, 2 km, 3 km, 4 km, 5 km, 6 km, 7 km, 8km, 9 km, 10 km, and 11 km, averaged for July–August 2011. Thin white lines indicate refers to the contour of pressure deviation, as defined in Fig. 2, and thick white lines highlight the Tibetan Plateau area.

[Figure]

**Figure A2.** EMAC simulated averaged daily deep convection events for each month in 2011. White lines highlight the Tibetan Plateau area.

**Reply to Anonymous Referee #2**

Referee comments are in black. Author responses are in blue.

**General comments:**

This interesting work studies the aerosol chemical composition in the Asian Tropopause Aerosol Layer (ATAL), and uses EMAC model running at a high resolution to investigate the links between ATAL and Asian Summer Monsoon (ASM). The aerosol properties in the ATAL have been the subject of discussion over the past decade and have received quite some attention recently. I agree with the Reviewer#1 that the results of this work nicely complement previous studies. The manuscript is well written, the methodology is detailed described and sound, and the results are well presented and discussed. The authors have addressed most of my concerns in the response to Reviewer#1. I feel this work is suitable for publication in ACP after address the following minor concerns.

We thank the anonymous referee for his/her insightful comments. Below are our point-by-point responses to referee's comments in detail.

**Specific comments:**

1) It would be best to rephase some expressions in the manuscript, considering the similarity index of 23% (see Similarity Report). No doubt regarding the originality of this work, which will be mentioned in my later comments, but I feel some re-wording to credit the previous works in a better way would help improve this manuscript.

The similarity report for our manuscript showed that the matching words resulted mostly from common terms, such as "*the upper troposphere and lower stratosphere*", "*over the Tibetan Plateau*" and "*the Asian summer monsoon (ASM)*", as well as from some phrases used to describe the model and methods. They were distributed over different paragraphs and pages. As suggested by the reviewer, we have strived as much as possible to rephrase these expressions so that the percentage of similarity can be dramatically reduced. Specific revisions can be found in the marked-up version of revised manuscript.

2) About the altitudes of 15-18 km. In order to avoid confusion, please find a suitable place to clearly state that whether it is above the sea level or over ground.

In the first paragraph of our manuscript (P2L9-10 of the ACPD version), we stated "*between 13-18 km above sea level (a.s.l., hereafter all altitudes are referred to a.s.l. except when specified differently)*". This statement appears to be not entirely sufficient for clarification. As suggested by the reviewer, we have added such note to the legends of related figures in the revised manuscript.

3) P5L24. I am wondering that how is organic aerosol formation simulated in the ORACLE sub-model, if partitioning of secondary organic aerosol between gas and particle phases is not considered? Some elaboration may be needed here.

Thanks to the referee, we have noticed that this text does not correspond to the simulation performed, as ORACLE was not used in our numerical experiment. We have rewritten the sentences in the revised manuscript as "*The primary OC aerosol is assumed to be emitted by 65% as hydrophilic and 35% as hydrophobic (Pringle et al., 2010). Following Pringle et al. (2010) and Pozzer et al. (2012), secondary OC particles are directly emitted as primary OC. The organic aerosol formation and chemical aging can be calculated by the ORACLE submodel in EMAC (Tsimpidi et al., 2014). However, in this work the partitioning of secondary organic compound between the gaseous and aerosol phases is not considered and the ORACLE module was not used in the simulation. The aging that leads the conversion of hydrophobic BC and OC to hydrophilic ones is realized by the collisions with hydrophilic particles and the condensation of inorganic gases such as sulfuric acid (Pringle et al., 2010 and references therein)*".

4) P10L19. The acronym 'SS' for sea spray. Please place it at the first time when 'sea spray' was used.

We have given up using the acronym 'SS' in the revised manuscript considering that sea spray (or sea salt) is not frequently discussed in the paper.

5) P10L32. The contribution of WASO and ALW to aerosol extinction is much higher in 2011 than in 2010/2012. Is it due to the pronounced Nabro eruption in 2011? This eruption may enhance the highly hygroscopic components in UTLS such as sulfate originated from SO2, and enlarge the contributions from WASO and ALW? Some discussion about the difference between 2011 and 2010/2012 would be interesting to see.

The distributions of ionic (hygroscopic) species, including $SO_4^{2-}$, $HSO_4^-$ and $H^+$, and the effect of the 2011 Nabro eruption are shown and discussed in Sect. 3.3. We have added the statements "*The pronounced Nabro eruption of $SO_2$ in 2011 can enhance sulfuric acid and sulfate aerosols in the UTLS, and increase the contributions of WASO and ALW to the aerosol in the UTLS, especially in the lower stratosphere and outside the ASM cyclone, as shown and discussed in the following section*" to the last paragraph of Sect. 3.2, and "*Due to the effect of the Nabro eruption, higher $SO_4^{2-}$, $HSO_4^-$ and $H^+$ concentrations are found in the year 2011 than in 2010 and 2012 over the ASM region and the whole northern hemisphere, most profoundly in the lower stratosphere*" to the third paragraph of Sect. 3.3 in the revised manuscript.

6) P11L11. The definition of 'Ca2+*' needs to be clarified. Cations of Na, K, Mg and Ca are accounted for as 'Ca2+*'. Here, you mean by mass, mol, or charge balancing? Furthermore, Na+ is also a typical tracer for sea salt. Drop sodium and just sum up calcium, magnesium and calcium. Would this be a better tracer for mineral dust?

As shown in Figs. S7, 8, and 9, all these ionic species are accounted for by charge balancing. We have rewritten the sentences as "*(4) for simplicity, $Na^+$, $K^+$ and $Mg^{2+}$*

*together with $Ca^{2+}$ in the mineral dust are accounted for as $Ca^{2+,*}$ in the equivalent amounts since $Ca^{2+}$ is a typical ionic tracer for mineral dust (Ma et al., 2013). Note that these four ionic species are also contained in sea salt and generally $Na^+$ is taken as a tracer for sea salt. However, mineral dust appears to be a dominant source of all these species in the UTLS over the Tibetan Plateau since they are well correlated with the dust plume and the concentrations of $Ca^{2+}$ are much higher than those of $Na^+$ there*".
The distribution patterns of $Ca^{2+,*}$ with $Na^+$ being dropped (Fig. A3) are similar to those with $Na^+$ being included (Fig. 9). So we still take $Na^+$ into account for $Ca^{2+,*}$ as the maximum of $Na^+$ concentrations in the UTLS over the Tibetan Plateau is also dominated by the contribution from mineral dust.

7) P13L27. Typo? 'ACM' change to ASM ?
Done.

8) Section 3.5. A nice discussion to figure out the source of dust in the UTLS by holographic scanning the transport pathway from surface to 18 km. My first suggestion is: would it be more generally representative to show the results of 2010/2012, but place the results of 2011 in supplement? Anyway, it would not change the results significantly, because 'transport patterns are the same in 2010 and 2012 as in 2011'; however, 2011 is a special year tagged by the pronounced Nabro eruption with potential impacts on climate the general circulation, which may lead to unnecessary doubt from audience about its representative. The second suggestion is: would it be possible to have some discussion about the source or transport pathway of WASO and ALW (or water vapor), which also considerably contribute to aerosol extinction coefficient in the ATAL?
According to the reviewer comments, we have used the results of 2012 for both Fig. 3 and Fig. 12, and have moved the results of 2011 to the Supplement of the revised manuscript. For the second suggestion of the reviewer, we have added Figures S11, S12 and 13 (and Figures S13, S14 and 14) and related discussions about the distributions and transport of water vapor (and ammonia) in the revised manuscript, as can be seen in the last two paragraphs of Sect. 3.5: "*Figures S11, S12 and 13 present the geographical distributions of specific humidity, $q_v$, at various altitudes from the surface to 18 km over the ASM region from our EMAC simulations, averaged for July–August in the years 2010, 2011 and 2012, respectively. It is shown that deep convection of moisture occurs over the southern edge of the Tibetan Plateau, with the maximum $q_v$ attained from 10– 11 km altitude upward. This is similar to the reanalysis results from previous studies, e.g., Xu et al. (2014), who found the most significantly relative enhanced $q_v$ at 250–300 hPa over the same area. The geographical position of the mineral dust maximum in the upper troposphere over the Tibetan Plateau is apparently different from that of the moisture maximum, indicating that different pathways for the transport of aerosols and their gaseous precursors from the lower troposphere to the UTLS over the Tibetan Plateau.*

*Figures S13, S14 and 14 present the geographical distributions of gaseous ammonia ($NH_3$) at various altitudes from the surface to 18 km over the ASM region from our*

*EMAC simulations, averaged for July–August in the years 2010, 2011 and 2013, respectively. In addition to northern India, which is located to the southwest of the Tibetan Plateau, an enhancement of surface $NH_3$ is also found in the central main body of the Tibetan Plateau, although the maximum $NH_3$ mixing ratio in the latter area (~10 ppbv) is about half of that in the former area (~20 ppbv). It is shown that surface ammonia can be more efficiently convected and transported to the UTLS from the Tibetan Plateau than from the Indian region, leading to high levels of $NH_3$, e.g., about 0.1 ppbv at 17 km altitude. Such large amounts of lofted ammonia accumulate within the ASM anticyclone, providing the basic gases favourable for the formation of nitrate aerosols (especially in the years 2010 and 2012 when the volcanic eruption effect is relatively small with respect to 2011), as discussed in Sect. 3.3.".*

The sources, formation mechanisms and transport pathways of aerosols in the ATAL are very complicated and relevant to various physiochemical, cloud and dynamical processes, which will be investigated thoroughly in the next study.

9) P17L9. I am curious that why finer resolution (T106L90) underestimate the convective vertical transport, but not the coarser resolution. Should the finer resolution represent convective processes better, although none of them explicitly describe the convective processes? Would you help me understand it.

As pointed out by the reviewer, the vertical velocity manifests on finer horizontal resolution, approaching to be more realistically representing the convective processes. On the other hand, cumulus convection is a sub-grid scale phenomenon in all GCMs and has to be parameterized using model's resolved variables. For the extended versions of the convective parameterization scheme at ECMWF developed by Tiedtke (1989) and Nordeng (1994), the large scale convective activity is related to the convective available potential energy (CAPE), and the cloud base mass flux ($M_B$) can be expressed as:

$$M_B = \frac{CAPE}{\tau} \left\{ \int_{base}^{top} \left[ \frac{(1+\delta\overline{q})}{C_p\overline{T}_v} \frac{\partial\overline{s}}{\partial z} + \delta \frac{\partial\overline{q}}{\partial z} \right] \eta \frac{g}{\rho} dz \right\}^{-1}$$

(see the paper of Nordeng (1994, equation 33) for detailed information about the variables and formula). $M_B$ is inversely proportional to the adjustment time scale $\tau$, giving too little convective mass flux when a too long adjustment time is chosen.

To estimate the cloud ensemble properties properly, among other parameters, the convection relaxation time $\tau$, which should be smaller (larger) with increasing (decreasing) horizontal resolution, also needs to be adjusted. Following Nordeng (1994), who suggested that as a rule of thumb the change in $\tau$ should be inverse proportional to the change in resolution, an algorithm similar to that used in the ECMWF model, i.e., $\tau[s] = \min(3 \cdot 3600, 2 \cdot 3600 \cdot 63/N)$, where N denotes the spectral resolution, is applied in EMAC. In this study, N was set to be 106 since we performed T106 simulations. However, such algorithm was tuned to T63 simulations and might lead to

biases (either positive or negative) when applying to other resolutions. Actually, Nordeng (1994, chapter 6.1) reports deviations from his rule of thumb. In going from T106 to T213, which is a doubling in resolution, he proposed according to his estimate a halving in $\tau$. But the change from T63 to T106, which increases resolution just by a factor of 1.67, also required a halving in $\tau$ in his model. We will try to improve the convection parameterizations for T106L90 simulations in the next study.

10) The conclusion. I feel add some clear statement to highlight the originality and novelty of this work would help audience get the full picture of this nice study. In the introduction and results discussion, authors made extensive comparisons with previous studies to evaluate the results of this study. Add some sentences to clearly state the new findings or improvements of this study will be helpful. For example, including nitrate which is missing in lots of previous studies, find more dust contribution by updating dust emission scheme and etc..

Thanks to the reviewer, we have added the following sentences to the end of the second last paragraph and the last paragraph of the conclusions in the revised manuscript, respectively: *"
[revised manuscript text omitted]

[Figure]

**Figure S11.** Same as Fig. 13, but for the year 2010.

[Figure]

**Figure S12.** Same as Fig. 13, but for the year 2011.

[Figure]

**Figure S13.** Same as Fig. 14, but for the year 2010.

[Figure]

**Figure S14.** Same as Fig. 14, but for the year 2011.